# Sparse Diffusion Autoencoder for Test-time Adapting Prediction of Complex Systems

**Jingwen Cheng**[*]
Department of Electronic Engineering
Tsinghua University
Beijing, China

**Ruikun Li**[*]
Shenzhen International Graduate School
Tsinghua University
Shenzhen, China

**Huandong Wang**[†]
Department of Electronic Engineering
BNRist, Tsinghua University
Beijing, China

**Yong Li**
Department of Electronic Engineering
BNRist, Tsinghua University
Beijing, China

## Abstract

Predicting the behavior of complex systems is critical in many scientific and engineering domains, and hinges on the model's ability to capture their underlying dynamics. Existing methods encode the intrinsic dynamics of high-dimensional observations through latent representations and predict autoregressively. However, these latent representations lose the inherent spatial structure of spatiotemporal dynamics, leading to the predictor's inability to effectively model spatial interactions and neglect emerging dynamics during long-term prediction. In this work, we propose SparseDiff, introducing a test-time adaptation strategy to dynamically update the encoding scheme to accommodate emergent spatiotemporal structures during the long-term evolution of the system. Specifically, we first design a codebook-based sparse encoder, which coarsens the continuous spatial domain into a sparse graph topology. Then, we employ a graph neural ordinary differential equation to model the dynamics and guide a diffusion decoder for reconstruction. SparseDiff autoregressively predicts the spatiotemporal evolution and adjust the sparse topological structure to adapt to emergent spatiotemporal patterns by adaptive re-encoding. Extensive evaluations on representative systems demonstrate that SparseDiff achieves an average prediction error reduction of 49.99% compared to baselines, requiring only 1% of the spatial resolution.

## 1 Introduction

The dynamics of complex systems are driven by the nonlinear interactions and co-evolution of numerous components, giving rise to rich emergent spatiotemporal structures, as seen in fluid dynamics [1], climate science [2], and molecular dynamics [3]. Accurate long-term prediction of these systems is crucial for many real-world applications [4, 5]. However, complex systems are typically observed in high-dimensional spaces with unknown intrinsic dynam-

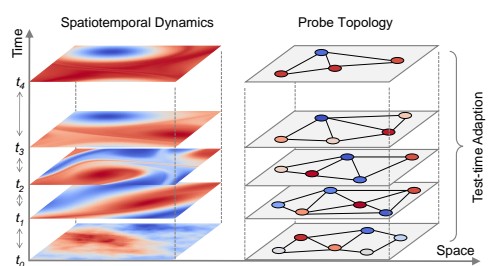

Figure 1: Adapting probe topologies.

---

[*]Equal contribution (see § 7).

[†]Corresponding author (wanghuandong@tsinghua.edu.cn)

39th Conference on Neural Information Processing Systems (NeurIPS 2025).

ics, making high-fidelity, long-term prediction a challenging and computationally expensive problem [6, 7, 8].

Data-driven methods offer an equation-free and computationally friendly approach. The core idea is to build reduced-order models of high-dimensional observations using encoder-decoders, which project observed states to a low-dimensional latent space for prediction, thereby improving computational efficiency [6, 9, 10, 11, 12]. However, most existing work employs parameterized neural network encoders that implicitly encode spatial structure by compressing observational spatial information into a latent representation vector. This conversion of spatially-correlated data into latent vectors hinders predictors from effectively understanding and modeling the system's inherent spatial interactions and structural relationships [5]. For instance, in fluid convection, heat drives fluid to form rising plumes and sinking cold flows. These specific spatial structures, their relative positions, and interactions are critical for understanding and predicting convection patterns [13]. Considering that spatial interaction patterns and emergent structures continuously change during long-term evolution, losing such information progressively amplifies the misalignment between the encoder-decoder and predictor in long-term predictions. This leads to a core question: Can we construct a reduced-order model that preserves crucial spatial structure and continuously adapts to newly emerging dynamics patterns during prediction?

This work is inspired by a recent finding that the system state in the entire spatial domain can be effectively reconstructed from sparse observation points [14, 15, 16]. The data disparity between sparse points and the full spatial grid strongly suggests a promising encoding paradigm: aggregating the full-space dynamic information onto a much smaller set of sparse probes [17]. These probes and their spatial topology constitute the dynamical skeleton of the spatiotemporal dynamics, as shown in Figure 1. In this paradigm, the model is capable of adapting to the system's latest spatiotemporal patterns during long-term predictions by dynamically adjusting probe positions and topology based on the re-encoded predicted states [6, 18]. Nevertheless, achieving effective probe-based aggregation and adaptive updating in practice faces two key challenges: 1) A lack of theoretical understanding for aggregating rich spatiotemporal dynamics from a continuous domain into a small set of discrete probes; 2) Frequent re-encoding during long-term prediction introduces significant computational overhead and potential accumulation of prediction errors.

To address these challenges, we propose a novel Sparse Diffusion Autoencoder, SparseDiff, consisting of a codebook-based sparse encoder and an unconditional diffusion decoder. The sparse encoder learns a representative pattern codebook from historical spatiotemporal trajectories, enabling it to dynamically aggregate and project full-space observational data onto a spatial topology formed by sparse probes, thus constructing a reduced-order model. On this probe topology, we model spatiotemporal dynamics using a graph neural ordinary differential equation and explicitly introduce a diffusion term to capture spatial interactions and information propagation among probes. Finally, we pad the probe predictions to the full spatial domain to serve as the initial state for the diffusion process, enabling rapid reconstruction of the full spatiotemporal field. SparseDiff's key innovation and strength lie in its test-time adaptation: during long-term prediction, it utilizes the learned codebook to re-encode the latest prediction, dynamically adjusting and constructing a probe topology better matched to the current system state to continuously adapt to emerging spatiotemporal patterns. Experimental evaluation on simulated and real-world systems demonstrates that SparseDiff outperforms baselines by over 49.99% in long-term predictions, and reliably predicts full-space spatiotemporal dynamics using less than 1% of grid points as probes.

The highlights of this work are summarized as follows:

- We propose SparseDiff, a novel autoencoder that uses sparse probes to construct a reduced-order model that effectively preserves the spatial structure of spatiotemporal dynamics.

- We introduce a test-time adaptation strategy via re-encoding, allowing the model to dynamically adapt its probe-based representation to emerging dynamics patterns, significantly improving long-term prediction accuracy and computational efficiency.

- Experiments demonstrate that SparseDiff significantly outperforms baselines in long-term prediction accuracy and achieves high efficiency by reliably predicting full-space dynamics using only approximately 1% of grid points as probes. Our code is open-source: `https://github.com/tsinghua-fib-lab/SparseDiff` .

## 2 Preliminary

### 2.1 Problem Definition

In this work, we focus on spatiotemporal dynamics $\frac{du}{dt} = f(u, x, t, \frac{\partial u}{\partial x}, \frac{\partial^2 u}{\partial x^2}, ...)$ on a 2D regular spatial domain $x \subset \mathbb{R}^2$. Such systems can often be formalized as time-dependent partial differential equations, for example, the Navier-Stokes equations, which include a Laplacian operator term acting on the spatial domain [19]. Without loss of generality, we address the problem of data-driven modeling of system dynamics from historical evolving trajectories $U = \{u(x, t)\}^T$, thereby extending our focus to real-world systems like climate dynamics where explicit equations are unknown. In practice, the spatial domain is discretized into an $h * w$ grid. Consequently, the full-domain observation state at each timestamp is represented as a tensor $u \in \mathbb{R}^{h \times w}$. Given a historical observation trajectory of *lookback* steps, we predict multiple future steps in an autoregressive manner, forecasting one step at a time.

### 2.2 Guided Diffusion for Sparse Reconstruction

Diffusion methods have recently been shown to reliably reconstruct the full spatial domain's system state, guided by sparse observations [16, 15, 14]. Diffusion models aim to learn a probabilistic mapping from a simple prior distribution, such as a standard Gaussian to a complex target data distribution [20, 21, 22]. This is achieved by defining a forward diffusion process that gradually adds noise to data instances and training a reverse process to sequentially denoise them. We represent the original data point as $x_0$. The forward stage transforms $x_0$ into a noisy version $x_n$ over $n$ steps, governed by the equation $x_n = \sqrt{\bar{a}_n} x_0 + \sqrt{1 - \bar{a}_n} \epsilon_n$, $\epsilon_n$ and $\{\bar{a}_n\}$ represent the Gaussian noise and noise schedule [23], respectively. The learned reverse diffusion process models the transition from noise back to data through a sequence of conditional distributions given by

$$p_\theta(x_{n-1}|x_n) := \mathcal{N}(x_{n-1}; \mu_\theta(x_n, n), \sigma_n^2 \mathbf{I}), \tag{1}$$

where $\mu_\theta = \frac{1}{\sqrt{\alpha_n}}(x_n - \frac{1 - \alpha_n}{\sqrt{1 - \bar{\alpha}_n}} \epsilon_\theta(x_n, n))$ and $\{\sigma_n\}$ are step dependent constants. The term $\epsilon_\theta$ represents the model's prediction of the added noise, typically implemented using a parameterized neural network architecture like a UNet or Transformer. The optimization of this network's parameters is performed by minimizing the objective function [23]

$$L_n = \mathbb{E}_{n, \epsilon_n, x_0} ||\epsilon_n - \epsilon_\theta(\sqrt{\bar{\alpha}_n} x_0 + \sqrt{1 - \bar{\alpha}_n} \epsilon_n, n)||^2. \tag{2}$$

This loss function is derived from the negative log-likelihood $\mathbb{E}_{x_0 \sim q(x_0)}[-p_\theta(x_0)]$. Once trained, the diffusion model progressively denoises from Gaussian noise to yield high-fidelity data samples. To guide diffusion in reconstructing the full spatial domain state from $K$ sparse observations $\mathcal{M}(x_K)$, Bayes' rule guides the diffusion gradient direction [24] to be

$$\nabla_{x_n} \log p(x_n|\mathcal{M}) \approx -\frac{1}{\sqrt{1 - \bar{\alpha}_n}} \epsilon_\theta - \zeta \nabla_{x_n} ||y - \mathcal{M}(x_K)||_2^2, \tag{3}$$

where $y$ represents the noise values at the sparse observation points and $\zeta = 1/\sigma^2$.

## 3 Methodology

In this section, we first introduce the proposed sparse diffusion autoencoder to discover the skeleton of spatiotemporal dynamics, namely the probe topology. Subsequently, we design a diffusion graph neural ordinary differential equation on the probe topology to model spatiotemporal dynamics. Finally, we propose the test-time adaptation strategy to dynamically sense emerging spatiotemporal patterns in long-term prediction. The overall framework is illustrated in Figure 2.

### 3.1 Sparse Diffusion Autoencoder

We introduce our approach for discovering the dynamical skeleton of complex systems, utilizing a codebook-based sparse encoder and a guided diffusion decoder.

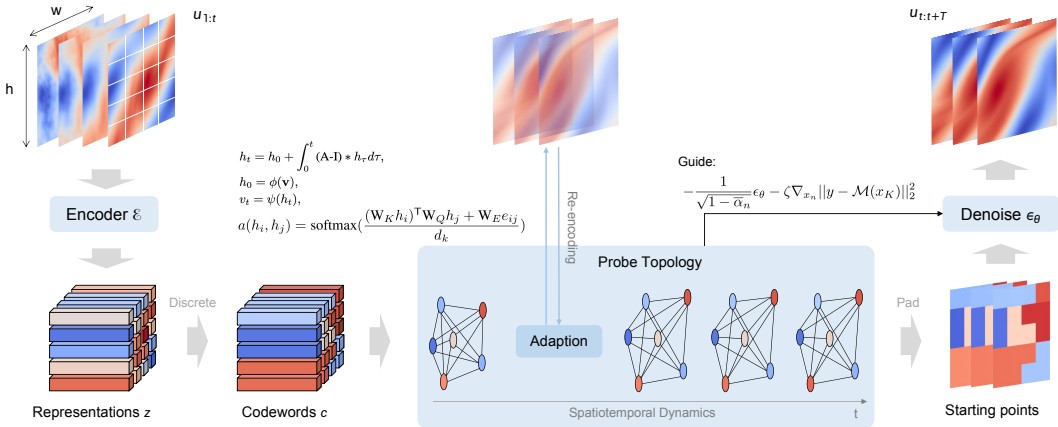

Figure 2: Overall framework of proposed Sparse Diffusion Autoencoder.

### 3.1.1 Codebook-based Sparse Encoder

To discover the skeleton of spatiotemporal dynamics on the spatial domain, we first identify and record the rich dynamical patterns inherent in complex systems. Specifically, we maintain a codebook $C$ containing $K$ codewords for spatiotemporal dynamics, where each codeword $c \in \mathbb{R}^d$ is a $d$-dimensional learnable vector. For a given historical observation $\mathbf{u} \in \mathbb{R}^{t \times h \times w}$, we employ an encoder $\mathcal{E}$ to encode along the time dimension, yielding a $d$-dimensional representation vector $z \in \mathbb{R}^{d \times h \times w}$ for the full-domain states. We replace the representation vector $z$ with the most similar codeword $c$, which serves as the decoder's input. The decoder then reconstructs the original observation $\hat{\mathbf{u}}$. The encoder-decoder and the codebook are trained by minimizing the objective function [25]

$$L = \log p(\mathbf{u}|c) + ||\text{sg}[\mathcal{E}(\mathbf{u})] - c||_2^2 + \beta||\mathcal{E}(\mathbf{u}) - \text{sg}[c]||_2^2, \tag{4}$$

where sg indicates gradient detachment. Through this process, we effectively record the rich spatiotemporal dynamical patterns from the observation trajectories within the pretrained codebook.

In the inference stage, we first discretize the historical observation $u$ into a set of $k$ hit codewords $\{c_i\}_{i=1}^k$ using pretrained encoder $\mathcal{E}$ and codebook $C$. We define the governing region of each hit codeword $c_i$ as the set of spatial grid points that are mapped to it, denoted as $u_{c_i}$. Consequently, the full spatial domain is divided into $k$ region types by the $k$ hit codewords (a single region type might consist of dispersed patches). These $k$ regions represent $k$ local-scale spatiotemporal units. Therefore, we accordingly construct probe set $\mathcal{V} = \{v_i \in \mathbb{R}^l\}_{i=1}^k$ to represent them, where $l$ is the lookback window size.

For probe $v_i$, we select one spatial grid point from its corresponding codeword's governing region $u_{c_i}$ as its coordinate. We use random selection here instead of averaging, ensuring that the probe falls within patches even if the patches of its region are spatially dispersed. Subsequently, we average the historical observation sequences of all grid points within $u_{c_i}$ to aggregate them as probe states $\mathbf{v_i} \in \mathbb{R}^l$. Finally, we connect all probes to construct the topological structure $\mathcal{G} = \{\mathcal{V}, E\}$, and assign different weights $e_{ij} \in E$ to each edge to characterize the spatial association strength. Specifically, edge weights $e_{ij}$ quantify the spatial association strength from probe $v_i$ to probe $v_j$. We consider the governing region $u_{c_i}$ associated with $v_i$ and spatial neighborhood of every grid point within $u_{c_i}$. We count how many grid points belonging to $v_j$'s governing region $u_{c_j}$ appear within the combined neighborhoods of all points in $u_{c_i}$. This count is normalized along all probes to determine the edge weight $e_{ij}$. Thereby, we obtain the probe topology $\mathcal{G}$ of the spatiotemporal dynamics.

### 3.1.2 Guided Diffusion Decoder

We pretrain an unconditional diffusion model to capture the spatiotemporal patterns of the system state $u$. Given the set of probes $\mathcal{V}$, we use these probe values $v_i$ as sparse observations to guide the reconstruction by the diffusion decoder. Existing works [16, 15, 14] typically directly substitute probe values as the sparse observation $\mathcal{M}(x_K)$ in Equation 3 to guide reconstruction. However, our approach differs in that we guide the governing regions $u_{c_i}$ corresponding to the probes. That is, our probes represent the dynamical information of the entire governing region, not just single grid points. Therefore, we use probe value $v_i$ to guide diffusion in reconstructing its governing region.

Specifically, we fill the probe value $v_i$ to each grid point within its governing region $u_{c_i}$, serving as $\mathcal{M}(x_K)$ in Equation 3 to guide reconstruction. Furthermore, considering the computationally intensive of the denoising process, we use the filled full-space state instead of Gaussian noise as the starting point for denoising to accelerate the diffusion sampling process.

## 3.2 Probe-graph Diffusive Predictor

Given the probe topology $\mathcal{G}$ and probe states $\mathbf{v_i}$, we now predict the spatiotemporal dynamics of the system. Specifically, we model this process as a neural diffusion process on the graph [26]

$$
\begin{aligned}
h_t &= h_0 + \int_0^t (\text{A-I}) * h_\tau d\tau, \\
h_0 &= \phi(\mathbf{v}), \\
v_t &= \psi(h_t),
\end{aligned}
$$

where $h_t$ refers to the representation at time $t$, $\phi$ and $\psi$ are the parameterized encoder and decoders, and A is the time-varying diffusion coefficient between probes, calculated by a parameterized attention network. To explicitly model the spatial association strength between probes, we introduce the edge weights into the calculation of the diffusion coefficient as

$$
a(h_i, h_j) = \text{softmax}(\frac{(\mathbf{W}_K h_i)^\mathsf{T} \mathbf{W}_Q h_j + \mathbf{W}_E e_{ij}}{d_k}),
$$

where $h_i$ denotes the representation of the $i$th probe, $\mathbf{W}_Q$, $\mathbf{W}_K$, and $\mathbf{W}_E$ are learnable matrices. We employ a multi-head attention mechanism $\text{A} = \frac{1}{k} \sum_k \text{A}^k$ to capture complex dynamical mechanisms.

## 3.3 Test-time Adapting Prediction

During long-term prediction, complex system dynamics exhibit continuous emergence of spatiotemporal patterns, implying time-varying dynamical spatial interactions. Therefore, we continuously update the predictor during testing to adapt to the latest spatiotemporal dynamics patterns. For a $T$-step prediction, the prediction window is divided into $N$ sub-windows $\{w_n\}^N$. We assume each window possesses specific spatial interactions, with a corresponding probe topology sequence $\{\mathcal{G}_{w_n}\}^N$. We update the probe topology at the beginning of each window through re-encoding. Specifically, at a window transition time, we decode the probe states back to full space. Following the method in Section 3.1.1, we then re-encode the probe topology $\mathcal{G}$ for the latest dynamics within the current window using encoder $\mathcal{E}$ and codebook $C$. This topology is updated again at the next transition time.

We design a dynamic update strategy to dynamically determine the window transition time. Instead of using fixed-length sub-windows, we continuously monitor whether the recent evolution of each probe remains well-aligned with its assigned codeword in the latent space of the encoder $\mathcal{E}$. Specifically, after each re-encoding, we obtain $k$ hit codewords $\{c_i\}_{i=1}^k$, where each $c_i \in \mathbb{R}^d$ corresponds to a governing region. For each codeword, a probe $v_i$ is randomly selected to represent its region and forms a probe graph $\mathcal{G}$ for prediction. At each prediction step, we compute the latent consistency score $\chi_t$ by encoding the past $T$-step trajectory of each probe $v_i$ and evaluating its cosine similarity with the associated codeword $c_i$:

$$
\chi_t = \frac{1}{k} \sum_{i=1}^k \cos\left(\mathcal{E}(v_i^{t-T:t}), c_i\right).
$$

When $\chi_t$ drops below a predefined threshold $\tau$, it indicates that many probe representations have drifted away from their original latent codewords, suggesting a mismatch between the current probe partition and the evolving system dynamics. In response, we decode the probe graph of the past $T$ steps back to the full space using the diffusion decoder and recompute the probe topology via re-encoding. This strategy enables the model to adaptively reallocate codewords and update the graph structure in accordance with the varying speed and patterns of spatiotemporal evolution.

Table 1: Average performance of the trajectories predicted from different initial conditions with standard deviation from 10 runs. The best results are highlighted in bold.

| Systems | Lambda-Omega | | Navier-Stokes | | Swift–Hohenberg | | Cylinder-Flow | | Real-world | |
|---|---|---|---|---|---|---|---|---|---|---|
| | RMSE $\times 10^{-2}$ | SSIM $\times 10^{-1}$ | RMSE $\times 10^{-2}$ | SSIM $\times 10^{-1}$ | RMSE $\times 10^{-2}$ | SSIM $\times 10^{-1}$ | RMSE $\times 10^{-2}$ | SSIM $\times 10^{-1}$ | RMSE $\times 10^{-2}$ | SSIM $\times 10^{-1}$ |
| ConvLSTM | 5.613 ± 0.557 | 9.224 ± 0.125 | 30.674 ± 2.704 | 4.696 ± 0.083 | 9.556 ± 0.059 | 9.399 ± 0.006 | 11.098 ± 0.330 | 5.265 ± 0.144 | 13.682 ± 2.777 | 7.288 ± 1.195 |
| FNO | 5.138 ± 0.557 | 9.170 ± 0.514 | 15.992 ± 1.062 | 7.262 ± 1.642 | 19.693 ± 5.591 | 9.614 ± 0.009 | 19.854 ± 25.248 | **7.583 ± 0.525** | 12.639 ± 3.878 | 6.775 ± 1.368 |
| UNet | 8.324 ± 0.497 | 8.014 ± 0.393 | 16.277 ± 1.582 | 7.092 ± 0.837 | 14.881 ± 2.789 | 9.206 ± 0.225 | 21.349 ± 4.903 | 6.203 ± 0.196 | 13.928 ± 2.855 | 6.328 ± 0.974 |
| G-LED | 6.506 ± 0.395 | 8.992 ± 0.396 | 12.334 ± 0.485 | 8.095 ± 0.963 | 8.214 ± 0.937 | 9.572 ± 0.211 | 10.021 ± 0.873 | 7.059 ± 0.291 | 10.304 ± 2.009 | 6.768 ± 0.539 |
| Ours | **2.912 ± 0.187** | **9.601 ± 0.218** | **11.130 ± 3.241** | **8.492 ± 0.793** | **7.628 ± 3.102** | **9.675 ± 0.192** | **8.544 ± 0.239** | 7.392 ± 0.171 | **7.957 ± 1.207** | **7.781 ± 1.034** |
| | 53.56% | 9.05% | 38.64% | 28.63% | 57.42% | 1.89% | 62.67% | 17.39% | 37.86% | 15.51% |

# 4 Experiments

In this section, we validate the accuracy and efficiency of SparseDiff on simulated PDE systems and real-world datasets. Furthermore, we evaluate its robustness and generalization ability and examine the specific contribution of its components through ablation studies.

## 4.1 Experimental Setup

We conduct experimental validation on four PDE systems, including: 1) Lambda-Omega; 2) Navier-Stokes; 3) Cylinder Flow; and 4) Swift–Hohenberg systems. They encompass complex diffusion effects, convection terms, and higher-order spatial interaction terms, among other nonlinear dynamic components. Additionally, we also evaluate SparseDiff's performance in real-world applications on an open-source climate record dataset [27]. All models are trained on trajectories with different initial conditions and predict long-term (more than 100 steps) on new trajectories. Details on the equations, data generation, and training settings are in Appendix A.

**Baselines**   We compare our method against a set of state-of-the-art spatiotemporal forecasting models, including operator learning methods, recurrent architectures, and generative frameworks. Specifically, we consider Fourier Neural Operator (FNO) [28], ConvLSTM [29], UNet [30], and G-LED [31]. A detailed description of each baseline is provided in Appendix C.

## 4.2 Main Results

**PDE systems**   Table 1 shows the long-term prediction performance of all models. SparseDiff achieves the optimal prediction quality in almost all scenarios. This indicates that SparseDiff's test-time adaptation enhances long-term prediction by timely sensing of new spatio-temporal dynamics. Furthermore, compared to other baselines, another characteristic of SparseDiff is that it performs dynamic prediction on the probe topology. Other methods predict in the original or uniformly downsampled grid space and, unlike SparseDiff, fail to discover the spatial interactions of complex systems and structure them into an explicit topology to enhance the predictor's representational capacity.

**Real-world dataset**   We use the SEVIR dataset [27] for real-world weather forecasting. Specifically, we select the GOES-16 Channel 09 ($6.9\mu m$) infrared imagery ir069, which captures mid-level water vapor and is widely used for storm tracking. Each frame covers a $384 \times 384$ km area at 2 km resolution, yielding a $192 \times 192$ grid, with a temporal resolution of 5 minutes. Experimental results are shown in Figure 3, where SparseDiff achieves the most faithful prediction of complex climate patterns with the lowest error. This indicates that the proposed probe-based encoding scheme is of great value for applications in real-world com-

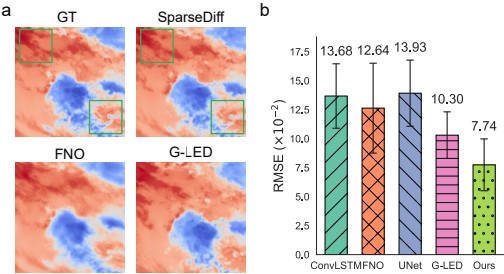

Figure 3: Real-world dataset. (a) Prediction visualization of different models. (b) RMSE comparison of different models.

plex systems, especially in large-scale observational data scenarios in geoscience. SparseDiff encodes

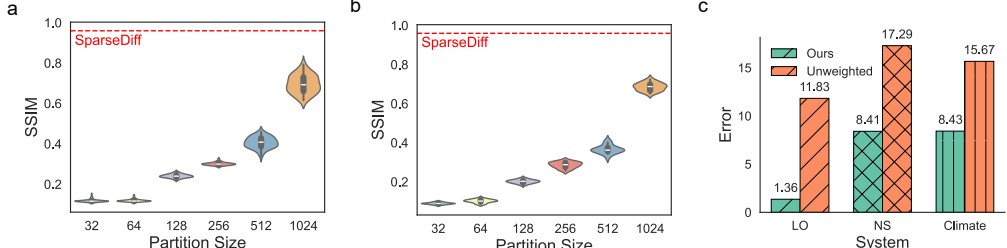

Figure 4: Ablation studies. Prediction performance with (a) uniform and (b) random probe selection. (c) Impact of probe topology edge weights on prediction.

high-dimensional observations into a sparse probe topology, thus reducing computational overhead while ensuring accuracy.

## 4.3 Ablation Study

We capture a compact low-dimensional skeleton of spatiotemporal dynamics through codebook learning, thereby improving prediction performance. The contrasting version is uniform downsampling or random selection of probes. We also establish heuristic calculation rules for the edge weights of the probe topology. The contrasting version is an unweighted graph (where all weights are set to 1). In the following, we ablate these two key components.

**Selection of Probes**  Probes are the most critical components in the SparseDiff framework, as they serve as the information carriers that represent the coarse-grained structure of the spatiotemporal dynamics. Their selection fundamentally determines the prediction quality of the entire model. Figures 4a and b illustrate the prediction performance of SparseDiff on the Swift–Hohenberg system when the probes are selected via spatially uniform and random sampling, respectively. We observe that even when the number of probes is increased to 1024, these two variants still perform significantly worse than our method. In contrast, the original version achieves high-quality predictions with as few as 150 probes. This highlights that our proposed codebook-based sparse encoder not only compresses the spatial domain effectively but also selects informative and dynamically representative probes. It demonstrates a strong ability to extract the low-dimensional spatiotemporal skeleton where the intrinsic dynamics reside.

**Probe-Graph Edge Weights**  When the edge weight feature is disabled, the edge-feature-aware attention mechanism described in Section 3.2 degenerates into standard inter-node attention that treats all probe connections equally. Without incorporating the spatial association strength $e_{ij}$, the model loses the ability to capture relative distances and directional relationships between probes, which are essential in spatiotemporal systems. As shown in Figure 4c, this leads to a notable performance drop on the Navier-Stokes system, where accurate modeling of spatial interactions is critical for capturing fine-grained dynamics.

## 4.4 Robustness

Here, we evaluate the robustness of SparseDiff to data observation noise and codebook size settings. Specifically, we first test the prediction performance with different preset codebook sizes on the Navier-Stokes system. The codebook size is the upper limit on the number of activated codewords. It determines the maximum number of probes SparseDiff can select to perceive spatiotemporal dynamics. Experimental results are shown in Figure 5a, where the prediction accuracy (SSIM) converges after the codebook size reaches 150. For the Navier-Stokes system with 128×128 resolution, this is

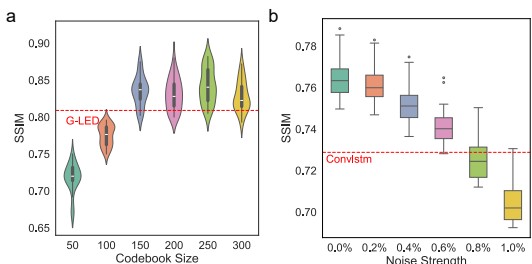

Figure 5: Robustness experiments. (a) Impact of codebook size on SparseDiff's performance on the Navier-Stokes system. (b) Impact of the noise on SparseDiff's performance on the real climate dataset.

quite a small number. This indicates that SparseDiff utilizes probes efficiently and achieves superior prediction with very low computational overhead.

Next, we examine SparseDiff's robustness to real-world noise. We apply Gaussian noise with varying percentage intensity to a real-world climate dataset (relative to the original data amplitude). Experimental results are shown in Figure 5b, where SparseDiff's performance deteriorates as the noise increases, but at moderate noise levels, it still outperforms the baseline without noise.

### 4.5 Generalization

We examine SparseDiff's out-of-distribution generalization ability. Specifically, using the Cylinder Flow system as the experimental subject, we evaluate SparseDiff's ability to predict flows with Reynolds numbers not included in the training set. For turbulent systems, the Reynolds number influences the viscous coefficient, thereby affecting collision frequency. For fluids with high Reynolds numbers, the flow exhibits chaotic tendencies and is thus difficult to predict over long periods. We train SparseDiff on Reynolds numbers less than 500 and predict on larger Reynolds numbers. Experimental results are shown in Figure 6, where the baseline shows rapid performance degradation in out-of-

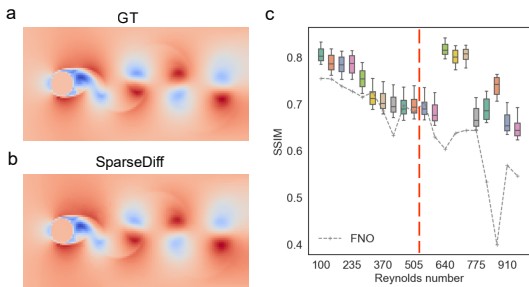

Figure 6: Generalization experiments. Visualized (a) ground truth and (b) SparseDiff prediction of turbulent at $Re = 660$. (c) SSIM as a function of Reynolds number.

distribution scenarios, whereas SparseDiff's performance consistently outperforms the baseline and exhibits fluctuations. The reason may be that a rich set of spatiotemporal dynamic patterns is recorded in SparseDiff's pretrained codebook. Although there are differences in the turbulent dynamic behaviors at different Reynolds numbers, local small-scale dynamic patterns share similarities and are thus recognized by SparseDiff and accurately generalized.

### 4.6 Trade-off between Accuracy & Efficiency

Here, we analyze how SparseDiff trades off between accuracy and efficiency. Specifically, during testing, SparseDiff can adjust the probe topology through re-encoding, thus constructing the most suitable form for newly emerging spatiotemporal dynamics. Frequent adjustments allow for sufficient perception of real-time dynamics, but the corresponding computational overhead increases. We conducted tests at different update intervals on the Navier-Stokes system, and the results are shown in Figure 7, where *Rollout Step* refers to the time index in the autoregressive prediction sequence, and each step

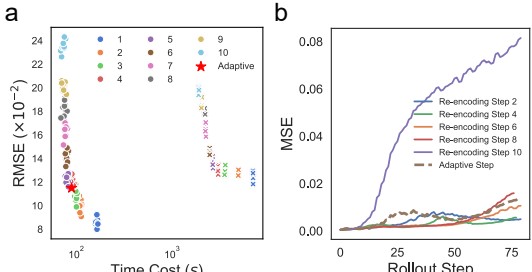

Figure 7: Trade-off of accuracy & efficiency. Circles represent SparseDiff, and crosses represent GLED.

corresponds to one forward prediction in the rollout process. According to Figure 7a, accuracy and time overhead exhibit a negative correlation.

We also compare with the best-performing baseline, G-LED. It can enhance long-term prediction accuracy by decoding and re-encoding at intervals, but its accuracy ceiling is lower than SparseDiff, and its time overhead is much greater than SparseDiff. Our adaptive re-encoding strategy proposed in Section 3.3 achieves a balance between accuracy and efficiency, as shown by the red star in Figure 7a.

### 4.7 Comparison of Graph Construction Schemes

To validate our proposed region-aware edge weighting scheme, we compare it against two primary baseline categories: (a) Learnable Graph Construction: The Graph Kernel Network (GKN) [32], a method designed for learnable graph construction; and (b) Heuristic Alternatives: Using the same GRAND predictor but replacing our learned $e_{ij}$ with static graph structures, including k-nearest

neighbors (kNN) based on Euclidean coordinates ($k = 20\%$ of probe count), and spectral clustering followed by kNN in the embedding space.

We report the average RMSE over a 100-step autoregressive prediction on the LO system. As shown in Table 2, our region-aware edge scheme yields significantly better performance than both the learnable GKN and the heuristic variants. This demonstrates that explicitly encoding region-level adjacency improves the expressiveness of the model for probe-based dynamics.

Table 2: Comparison of different graph construction methods.

| Method | RMSE $\times 10^{-2}$ |
|---|---|
| Ours | 2.873 |
| GKN | 4.210 |
| GRAND + kNN | 3.287 |
| GRAND + spectral | 3.019 |

## 5 Related Work

### 5.1 Encoding Dynamics of Complex Systems

Discovering the low-dimensional latent space containing the intrinsic dynamics of complex systems represents a core challenge in long-term prediction. To accelerate spatiotemporal dynamical system prediction, some earlier studies focus on approximating solutions on coarse grids. Lee et al. [7] utilize Gaussian processes and diffusion maps to coarse-grain high-resolution microscopic observations into coarse-scale PDEs. Bar-Sinai et al. [33] employ uniform grid downsampling of the continuous spatial domain for specific nonlinear partial differential equations. This coarse-grid approach has inspired several recent works [31, 34]. With advancements in representation learning, autoencoders have been employed to data-drivenly uncover the low-dimensional latent space of high-dimensional observations [6, 35]. Prediction models such as recurrent neural networks [6] and neural operators [9] operate directly within the low-dimensional space constructed by autoencoders. To ensure the latent space aligns with physical intuition, existing research suggests integrating autoencoders with physical priors [36]. Li et al. [11] restrict the output semantics of the encoder by specifying timescales and intrinsic dimensions. Wu et al. [10] combine delay embeddings with feature embeddings to discover the low-dimensional manifold of high-dimensional observations. Additionally, some work [37] learns the latent coordinates of partial differential equations through autoencoders to reveal governing equations. In contrast to these methods, our proposed probe topology adaptively senses spatial structure, without enforcing a regular grid. This explicitly preserves the tight spatial correlations of spatiotemporal dynamics with a high compression ratio.

### 5.2 Generative Modeling for Complex Systems

Diffusion models have performed remarkably well in generative tasks, with their significant achievements in video synthesis [38, 39] and time series modeling [40, 41] inspiring numerous studies in complex system dynamics prediction. Some works [42, 15] have utilized large-scale pre-trained diffusion models to reconstruct high-fidelity data from lower-fidelity samples or sparse measurement data. Physics knowledge is also integrated. For example, known partial differential equations provide physical constraints (PDE loss) for the denoising step, improving accuracy [43]. Beyond reconstruction, diffusion models have also been applied to generate specific dynamical data. Li et al. [44] propose a machine learning approach for generating single-particle trajectory data in high Reynolds number three-dimensional turbulence. Lienen et al. [13] treat turbulence simulation itself as a generative task, employing diffusion models to capture the distribution of turbulence induced by unseen objects and generate high-quality samples for downstream applications. Other strategies focus on directly embedding dynamics or prediction processes into the diffusion mechanism. Cachay et al. [45] align the temporal evolution axis of spatiotemporal dynamics with diffusion process steps, replacing noise injection with temporal interpolation and the denoising operation with prediction. G-LED [31], a more direct approach, incorporates system states as prediction targets and utilizes predicted future frames as conditions to guide the diffusion model in reconstructing the original high-fidelity states. Li et al. [5] embed multiscale features as conditioning for Diffusion modeling, while [46] dynamically adjust the number of denoising steps based on dynamical timestamps. Compared to these methods, we introduce a novel sparse encoder working in tandem with a diffusion decoder. This combination not only reveals the low-dimensional spatial structure of long-term dynamics but also enables test-time adaptation to emerging spatiotemporal dynamics.

# 6    Conclusion

Modeling the long-term dynamics of complex systems is a challenging problem. Data-driven methods are often constrained by the difficulty in explicitly preserving the inherent spatial interactions of spatiotemporal dynamics in latent vectors. This information loss is exacerbated by new patterns that continuously emerge during the system's long-term evolution. Therefore, we propose a novel Sparse Diffusion Autoencoder, SparseDiff, a prediction framework that preserves the system's spatial structure and adapts at test-time. SparseDiff coarsens the full spatial domain state into a probe topology by means of discrete codebook learning to construct a compact skeleton of spatiotemporal dynamics. The pretrained codebook is able to capture the rich dynamic patterns of complex systems. Once training is complete, SparseDiff can adjust the probe topology to adapt to emerging dynamic patterns during testing and without updating weights. On the probe topology, we design an edge-weight-aware graph neural diffusion ordinary differential equation to model spatiotemporal dynamics, thereby predicting the future states of the probes and guiding the diffusion model to efficiently reconstruct back to the full spatial domain. Experiments on simulated and real-world systems show that SparseDiff can achieve optimal long-term prediction and exhibits excellent robustness and generalization ability.

**Limitation & Future Work**    SparseDiff's spatiotemporal dynamics module relies on the perception of edge weights, therefore the rules for calculating weights will affect its long-term prediction quality. Furthermore, the state initialization of probes is achieved through average aggregation, which may lose high-frequency information at a few grid points. Future research will focus on end-to-end graph structure learning and probe initialization strategies.

# 7    Author Contributions

Ruikun Li, Huandong Wang and Yong Li launched this research and provided the research outline. Ruikun Li, Jingwen Cheng and Huandong Wang designed the research methods. Jingwen Cheng performed the experiments and prepared the figures & technical appendix. Ruikun Li wrote the original manuscript. Huandong Wang and Yong Li provided critical revisions.

# Acknowledgements

This work was supported in part by the National Natural Science Foundation of China under grant 62171260 and 92270114.

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

## Impact Statement

This paper presents work whose goal is to advance the field of Machine Learning. There are many potential societal consequences of our work, none which we feel must be specifically highlighted here.

## A   Data Generation

Below, we provide an overview of the dynamics and data generation process for each complex system. **Lambda–Omega (LO) system** is governed by

$$\begin{cases} \dot{u}_t = \mu_u \Delta u + (1 - u^2 - v^2)u + \beta(u^2 + v^2)v \\ \dot{v}_t = \mu_v \Delta v + (1 - u^2 - v^2)v - \beta(u^2 + v^2)u, \end{cases} \tag{5}$$

where $\Delta$ is the Laplacian operator. Here, we set $\mu_v$ and $\mu_v$ to 0.5, while $\beta$ is 1.0.

**Navier-Stokes (NS) system** is governed by:

$$\dot{\omega}_t + (\mathbf{u} \cdot \nabla)\omega = \nu \Delta \omega + f, \tag{6}$$

$$\nabla^2 \psi = -\omega, \tag{7}$$

$$\mathbf{u} = (u, v) = \left( \frac{\partial \psi}{\partial y}, -\frac{\partial \psi}{\partial x} \right), \tag{8}$$

$$\omega = (\nabla \times \mathbf{u}) \cdot \hat{\mathbf{z}} = \frac{\partial v}{\partial x} - \frac{\partial u}{\partial y}, \tag{9}$$

where $f = A\left(\sin\left(2\pi(x + y + s)\right) + \cos\left(2\pi(x + y + s)\right)\right)$ is the driving force, and $\nu$ is the viscosity coefficient. Parameter values used in this simulation are set as follows: $\nu = 1.0$, forcing amplitude $A = 0.1$, and phase shift $s = 0$.

**Swift-Hohenberg (SH) system** is simulated as:

$$\frac{\partial u}{\partial t} = ru - 2\Delta u - \Delta^2 u + gu^2 - u^3, \tag{10}$$

where $r$ is the linear instability parameter, $g$ controls the strength of the quadratic nonlinearity, and $\Delta^2$ denotes the biharmonic operator. In this system we set $r$ to 0.7 and $g$ 1.0 separately.

**Cylinder flow (CY) system** is governed by:

$$\begin{cases} \dot{u}_t = -u \cdot \nabla u - \dfrac{1}{\alpha}\nabla p + \dfrac{\beta}{\alpha}\Delta u, \\ \dot{v}_t = -v \cdot \nabla v + \dfrac{1}{\alpha}\nabla p - \dfrac{\beta}{\alpha}\Delta v. \end{cases} \tag{11}$$

where $\alpha$ is set to 1.0, and $\beta$ corresponds to the dynamic viscosity $\mu$ as defined in Equation 12.

The systems under study are simulated across a diverse range of initial conditions. Temporal downsampling by a factor of 10 and 25 are applied to the LO and NS systems separately to reduce redundancy, while all the systems are spatially rescaled to a uniform $128 \times 128$ resolution.

The Cylinder flow system is modeled via the lattice Boltzmann method (LBM) [6], capturing complex vortex shedding phenomena governed by the Navier-Stokes equations in the presence of a cylindrical obstacle. The simulation operates on a lattice velocity grid, where relaxation dynamics are dictated by the kinematic viscosity and the Reynolds number. To ensure data quality, we begin recording after the flow reaches a statistically steady turbulent regime. The outputs are resampled in time by a factor of 300 and spatially interpolated to a $128 \times 64$ grid. We generate 50 training and 20 testing trajectories across varying flow conditions, with Reynolds numbers uniformly sampled: 10 training samples in the range Re $\in [100, 500]$ and 10 out-of-distribution (OOD) samples in Re $\in [500, 1000]$. Based on the relation

$$\mu = \frac{\rho U_m D}{\text{Re}} \tag{12}$$

where $\rho = 1$, $U_m = 0.08$, and $D = 0.2$, the corresponding dynamic viscosity spans $\mu \in [3.2 \times 10^{-5}, 1.6 \times 10^{-4}]$ for training and $\mu \in [1.6 \times 10^{-5}, 3.2 \times 10^{-5}]$ for OOD cases.

To standardize input scales and facilitate stable training, we apply min-max normalization independently across all channels.

# B   Model Architecture

Our model is composed of three key modules: a Codebook-based Sparse Encoder for spatiotemporal discretization, a Probe-graph Diffusive Predictor for latent dynamics modeling, and a Guided Diffusion Decoder for reconstructing full-resolution system states. Below, we detail the architectural settings used in each component.

```python
# --- Sparse Encoder ---
hidden_dim = 1024       # hidden dimension of MLP
embedding_dim = 512      # Latent dimension (d)
num_embeddings = M      # Codebook size, hyperparameter
# --- Diffusive Predictor ---
input_steps = 10         # number of lookback steps
feature_dim = 256        # Feature dimension
num_heads = 8            # Attention heads for calculating Matrix A
ODE_method = 'rk4'       # Numerical solver for ODE integration
# --- Unconditioned Diffision ---
n_channels = 128          # Base number of channels
ch_mults = [1, 2, 2]    # Channel multiplier for each resolution level
is_attn = [False, False, True]  # Whether to apply self-attention
dropout = 0.1             # Dropout rate in residual blocks
n_blocks = 2             # Number of residual blocks per resolution level
```

Table 3: Trainable parameter counts (in millions).

| Model | SparseDiff | | | FNO | ConvLSTM | UNet | G-LED |
|---|---|---|---|---|---|---|---|
| | Encoder | Predictor | Diffusion | | | | |
| Params (M) | $2.3 \times 10^{-2}$ | 1.32 | 25.8 | 23.90 | 5.47 | 10.91 | 26.3 |

# C   Baseline Implementation

We provide a brief description of the baseline methods used for comparison in our experiments. These methods represent a diverse set of state-of-the-art approaches for modeling spatiotemporal dynamics. Their corresponding trainable parameter counts are summarized in Table 3.

- **FNO** [28]: Fourier Neural Operator leverages fast Fourier transforms to model spatially continuous operators, enabling efficient learning of solution mappings for partial differential equations. It is widely adopted for learning complex physical dynamics.

- **ConvLSTM** [29]: ConvLSTM integrates convolutional structures into recurrent networks, allowing spatial correlations to be preserved while capturing temporal dependencies. It is a standard baseline for video and sequence-based spatial forecasting tasks.

- **UNet** [30]: UNet employs an encoder-decoder structure with skip connections to effectively combine global context and local details. It is particularly suitable for dense prediction tasks involving structured outputs.

- **G-LED** [31]: G-LED is a generative latent evolution model that models temporal dynamics using autoregressive attention in latent space, and reconstructs full-resolution outputs using a Bayesian diffusion model. It achieves strong performance on high-dimensional physical systems.

# D    Additional Results

## D.1    Application to Irregular or Sparse Spatial Domain

While our current implementation operates on 2D regular grids, SparseDiff can be applied to irregular or sparse spatial data by first transforming the inputs into a regular grid through interpolation or zero-filling, enabling processing within the same framework.

To validate this, we conduct two experiments:

**a. Sparse sampling on Navier-Stokes system:**    On the Navier-Stokes system, we randomly sample only 10% of grid points. These sparse samples are then interpolated onto a $128 \times 128$ regular grid before being fed into SparseDiff. Despite the sparsity, SparseDiff outperforms baselines that are given full-resolution inputs, as shown in Table 4.

Table 4: Sparse sampling on the Navier-Stokes system. We report average metrics of 100 prediction steps.

| Method | RMSE $\times 10^{-2}$ ↓ | SSIM $\times 10^{-1}$ ↑ |
|---|---|---|
| FNO (full grid) | 15.99 | 7.26 |
| G-LED (full grid) | 12.33 | 8.10 |
| SparseDiff (full grid) | 11.39 | 8.36 |
| SparseDiff (10% observations) | 12.27 | 8.19 |

**b. Irregular 2D wave equation dataset [47]:**    We also evaluate SparseDiff on the 2D wave equation with observations sampled on irregular meshes. These inputs are first completed onto regular grids using interpolation within observed regions and zero-filling in excluded regions before being processed by the model. Compared to FNO, SparseDiff maintains stable prediction accuracy (Table 5).

Table 5: Results on the irregular 2D wave equation dataset [47] of 100 prediction steps.

| Method | RMSE $\times 10^{-2}$ ↓ | SSIM $\times 10^{-1}$ ↑ |
|---|---|---|
| FNO | 23.76 | 7.81 |
| SparseDiff | 15.58 | 8.25 |

While these results demonstrate practical applicability to irregular or sparse inputs, we acknowledge certain limitations of this approach:

- For highly non-uniform spatial distributions, interpolating to a uniform grid can be problematic. Also, the sparse observation may be too under-sampled to allow reliable interpolation, leading to unnecessary computational cost or significant reconstruction errors.

- If the distribution of spatial inputs shifts between training and testing, performance may degrade, as our current model lacks coordinate-querying capabilities (unlike operator-learning models such as neural operators), and cannot generalize across arbitrary spatial layouts.

Nevertheless, the core design of SparseDiff is not inherently tied to regular grids. The key innovation of SparseDiff lies in its latent probe representation, which aggregates and models regional dynamics through sparse, learnable units. This representation is inherently flexible and does not require regular spatial structures.

The main constraint stems from the UNet-based diffusion decoder, which requires regular-grid input. However, this is a design choice rather than a fundamental limitation of the method. In future work, the diffusion decoder could be replaced with architectures that naturally support irregular domains, such as Graph Neural Networks. Alternatively, it could be substituted with other super-resolution models, including transformer-based approaches and operator-learning models, to further extend SparseDiff to irregular spatial settings.

## D.2 Visualization

We provide long-term prediction visualizations on three representative systems: NS, SH, and CY. For each case, the left column shows the ground truth, the middle column presents the prediction reconstructed via the diffusion decoder, and the right column shows the vanilla reconstruction by directly filling each codeword region with its corresponding probe value.

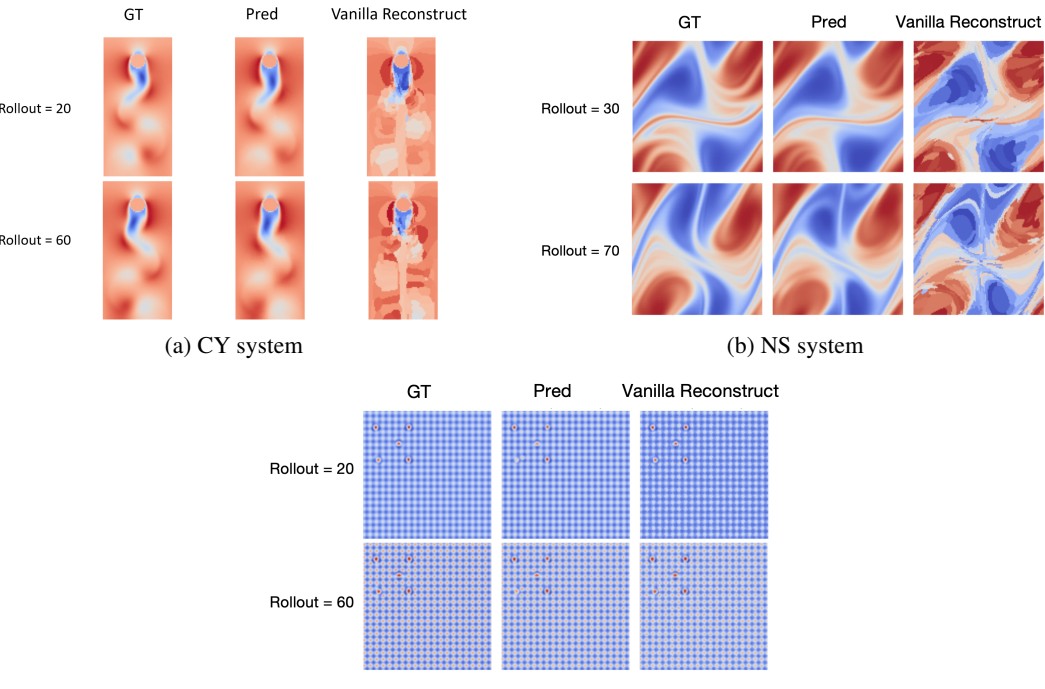

Figure 8: Long-term prediction results on three representative systems.

