# OpenReview forum: "Sparse Diffusion Autoencoder for Test-time Adapting Prediction of Complex Systems"
_NeurIPS.cc/2025/Conference — NeurIPS 2025 poster_

### Official Review · Reviewer_WcE9 · 2025-06-14

**Clarity:** 2
**Significance:** 3
**Originality:** 3
**Rating:** 3
**Confidence:** 3

**Summary:**

The paper “Sparse Diffusion Autoencoder for Test-time Adapting Prediction of Complex Systems” introduces SparseDiff, a framework for long-term forecasting of high-dimensional spatiotemporal dynamics (e.g., fluid flow, climate fields). SparseDiff compresses each high-dimensional frame into a handful of learned “probes” with a codebook-based sparse encoder, advances those probes through an edge-aware graph neural ODE, and then uses a guided diffusion decoder to regenerate the full field, thus preserving spatial structure while working in an extremely low-dimensional space.  Evaluated on PDE benchmarks and the SEVIR weather dataset, this test-time adaptation cuts long-range RMSE by roughly 50 % versus state-of-the-art baselines while requiring fewer than 1 % of grid points and remaining robust to noise and out-of-distribution Reynolds numbers.

**Questions:**

Please see the “Weaknesses” section. If the authors can supply adequate theoretical justification, I would be willing to consider raising my score.

**Ethical Concerns:**

["NO or VERY MINOR ethics concerns only"]

**Limitations:**

Please see the “Weaknesses” section.

**Paper Formatting Concerns:**

None found.

**Quality:**

3

**Strengths And Weaknesses:**

Strengths:

1. Robustness and generalization: maintains performance under noisy inputs and extrapolates to unseen Reynolds numbers without retraining.

2. Test-time adaptive encoding automatically refreshes the probe graph during inference, preventing drift in very long forecasts and preserving accuracy.

3. State-of-the-art long-horizon accuracy, reducing RMSE by about 50 % versus FNO, UNet and other strong baselines on four PDEs and a weather dataset.

Weaknesses:

1.  The introduction points out “a lack of theoretical understanding for aggregating rich spatiotemporal dynamics from a continuous domain into a small set of discrete probes”; yet the paper itself provides no such theoretical analysis—most notably, it does not quantify how prediction accuracy scales with the forecast horizon t, leaving readers without guidance on the temporal range over which the method’s outputs remain reliable.

2. The Methods section introduces a “Sparse Encoder,” yet it never defines what “sparse” specifically refers to—is it merely the step of replacing the latent vector z with its nearest codeword c before decoding? In addition, the paper offers no theoretical justification for why sparsity is needed or how it benefits the model.

3. In the PDE context $
h_t = h_0 + \int_{0}^{t} A\,h_t \, dt,
$, we also care about estimating and inferring the parameter A, as it is crucial for stability analysis and control design. Could your method be extended to provide reliable inference of A, rather than focusing solely on long-term state prediction?

---

> ### Author Rebuttal · Authors · 2025-07-31
>
> > ***Weakness 1: The introduction points out “a lack of theoretical understanding for aggregating rich spatiotemporal dynamics from a continuous domain into a small set of discrete probes”... it does not quantify how prediction accuracy scales with the forecast horizon t...***
>
> We sincerely thank the reviewer for raising this important question. We provide additional clarification from both a theoretical and empirical perspective.
>
> **a. Theoretical Support:**
>
> Many spatiotemporal dynamical systems, such as those in fluid dynamics or reaction-diffusion processes, evolve over time along a low-dimensional manifold after an initial transient phase. This has been formalized under slow manifold and center manifold theory [1, 2], and further supported by recent works that show data-driven models can recover effective latent dynamics in reduced spaces [3, 4].
>
> While these manifolds capture dominant macroscopic behavior, it has also been shown that global system states can be approximated from a sparse set of spatial observations, as analysed in [5]. Combining these insights, we construct a sparse latent representation by placing probes along the system’s manifold structure to efficiently capture and propagate global dynamics.
>
> Our probe topology serves this purpose by selecting informative locations and modeling their evolution through a graph neural network, allowing the model to track long-term dynamics over a compact, structure-aware representation. This aligns with the principles of modal compression and reduced-order modeling, ensuring both physical interpretability and computational efficiency.
>
> **b. Empirical Evidence: Fixed topologies rapidly lose expressiveness during long-term prediction**
>
> SparseDiff dynamically updates the probe topology by perceiving recent states within the previous forecast horizon. Therefore, a smaller forecast horizon implies more frequent updates. As shown in Figure 7(a), we evaluate how prediction accuracy evolves with different re-encoding intervals in the Navier-Stokes experiment. When the probe topology is updated at every step, the model maintains stable and accurate long-term predictions. As the interval increases, prediction error grows more rapidly. At every 10 steps, the MSE exceeds acceptable levels, indicating accuracy degradation.
>
> We further test a fixed-topology variant where no updates are performed throughout prediction. In this limiting case, the MSE quickly rises to ~100 within 50 steps, showing that the model fails to track system evolution. This underscores the importance of regularly updating the macroscopic structure.
>
> For reference, we report G-LED’s MSE over increasing horizons. As shown in the table below, G-LED accumulates errors faster and performs worse overall than SparseDiff.
>
> | Forecast Horizon (steps) | G-LED | SparseDiff (Adaptive)|
> | ------------------------ | ---------- |---------- |
> | 20                       | 0.012      |0.004      |
> | 40                       | 0.026      |0.007      |
> | 60                       | 0.039      |0.005      |
> | 80                       | 0.062      |0.011      |
>
> We will include the full G-LED error curves in the camera-ready version as a comparative reference to our method.
>
> > ***Weakness 2: The Methods section introduces a “Sparse Encoder,” yet it never defines what “sparse” specifically refers to... In addition, the paper offers no theoretical justification for why sparsity is needed or how it benefits the model.***
>
> To avoid ambiguity, we clarify the definition of the Sparse Encoder and the motivations for introducing sparsity into the representation.
>
> The term “Sparse Encoder” in our paper refers not merely to the step of replacing the continuous latent vector $\mathbf{z}$ with its nearest codeword $\mathbf{c}$ before decoding. More fundamentally, it denotes the use of a finite codebook to compress the system’s dynamic representations into a discrete set of sparsely activated latent patterns. Specifically, the encoder outputs a continuous vector $\mathbf{z}_i$, which is then discretized by mapping it to its nearest codeword $\mathbf{c}_j \in \mathcal{C}$, where $\mathcal{C}$ is a codebook with a predefined size $K$. Since every latent vector is constrained to fall within this finite set, only a small subset of representative modes is activated across the representation space, thereby introducing structured sparsity into the model.
>
> This sparse design is motivated by two key principles for modeling high-dimensional spatiotemporal systems:
>
> - Computational Efficiency and the Manifold Theory.
>
>   High-dimensional PDE systems are computationally expensive to model directly. However, their intrinsic dynamics often evolve on a much lower-dimensional manifold, an idea formalized in center manifold theory [1, 3, 4, 8]. Our sparse probe topology is explicitly designed to identify and approximate this low-dimensional manifold. This allows the predictor to operate in a highly efficient, reduced-order space that still captures the system's core dynamics.
>
> - Robust Long-term Prediction.
>
>   Sparsity is crucial for capturing emergent patterns in long-term forecasting. By forcing the model to represent the system state using a few key probes, the encoder must learn to capture the most essential dynamical components while filtering out redundant or noisy information. This focus on the dynamical skeleton makes the predictor more robust and better able to adapt to new structures as they emerge over time [8].
>
> > ***Weakness 3: In the PDE context... we also care about estimating and inferring the parameter A... Could your method be extended to provide reliable inference of A, rather than focusing solely on long-term state prediction?***
>
> While our primary objective is long-term trajectory prediction, the diffusion operator $A$ is not purely learned from data. It is explicitly designed based on physically inspired priors that reflect the spatial organization of the system.
>
> Specifically, our predictor is based on Graph Neural Diffusion (GRAND) [1], which models information propagation on a graph as a continuous diffusion process:
>
> $$
> h_t = h_0 + \int_0^t A  h_\tau \, d\tau,
> $$
>
> where $A$ defines the diffusion strength, and each $a_{ij}$ encodes how much node $j$ contributes to node $i$’s evolution over virtual time.
>
> In the original GRAND framework, diffusivity $a_{ij}$ is modeled purely through an attention mechanism over learnable probe features. In contrast, our approach represents each probe as a region-aggregated feature, computed by averaging the grid points within the corresponding codeword region $u_{c_i}$. Thus, we argue that similarity between probes should account not only for feature embeddings but also for how region-level adjacencies affect spatial interactions.
>
> To enhance the physical interpretability and spatial accuracy of the graph, we introduce an edge weight term $e_{ij}$ to measure the spatial adjacency between probe regions. Specifically, $e_{ij}$ quantifies the fraction of neighboring points in $u_{c_i}$ adjacent to those in $u_{c_j}$. This term is integrated into the attention-based diffusivity calculation:
>
> $$
> a(h_i, h_j) = \text{softmax}\left( \frac{(W_K h_i)^T W_Q h_j + W_E e_{ij}}{d_k} \right),
> $$
>
>
>
> where $W_E$ is a learnable weight matrix. This allows $e_{ij}$ to be an input feature, not a fixed multiplicative factor, ensuring the final diffusion strength $a_{ij}$ remains trainable via backpropagation. This design incorporates domain knowledge into the construction of $A$, reflecting the physical meaning of a diffusion operator rather than being arbitrarily fitted for improved prediction.
>
> To validate that our operator $A$ captures meaningful, physically grounded structure, we perform an ablation study comparing different graph construction methods, each defining a distinct form of $A$.
>
> Specifically, we compare:
>
> a. **Graph Kernel Network** [4], which learns the connectivity pattern end-to-end from data;
> b. **Heuristic alternatives** using the same GRAND predictor but replacing $e_{ij}$ with:
>
> * K-nearest neighbors based on Euclidean coordinates ($k = 20\%$ of probes);
> * Spectral clustering followed by KNN in the embedding space.
>
> The results, reported as average RMSE over 100-step prediction on the LO system, are shown below:
>
> | Method                                 | RMSE × 10⁻² |
> |----------------------------------------|--------|
> | Ours    |    2.873    |
> | GKN      |   4.210       |
> | GRAND + kNN            |   3.287     |
> | GRAND + spectral       |   3.019     |
>
> In summary, while our main goal is trajectory prediction, the probe graph defines a physically meaningful approximation of $A$. Instead of brute-force fitting, it utilizes spatial and latent structure. Empirical results show it outperforms both heuristic and learned methods, highlighting its potential for broader applications beyond prediction.
>
> ## Reference
> [1] E Knobloch, K Wiesenfeld, Bifurcations in fluctuating systems: The center-manifold approach.
>
> [2] N Fenichel, Geometric singular perturbation theory for ordinary differential equations.
>
> [3] Vlachas, P. R., et. al. Multiscale simulations of complex systems by learning their effective dynamics.
>
> [4] Floryan, D., & Graham, M. D. (2022). Data-driven discovery of intrinsic dynamics.
>
> [5] Brunton, B. W., et. al. Sparse sensor placement optimization for classification.
>
> [6] Chamberlain, B. P., et. al. GRAND: Graph neural diffusion.
>
> [7] Li, Z., Kovachki, et. al. Neural operator: Graph kernel network for partial differential equations.
>
> [8] Li, R., Wang, H., et. al. Predicting Long-term Dynamics of Complex Networks via Identifying Skeleton in Hyperbolic Space.

---

### Official Review · Reviewer_E8yK · 2025-06-29

**Clarity:** 3
**Significance:** 3
**Originality:** 3
**Rating:** 4
**Confidence:** 3

**Summary:**

The paper presents SparseDiff, a novel autoencoder framework for predicting complex spatiotemporal dynamics. The model first encodes the system's state into a sparse probe topology using a learned codebook of dynamic patterns. It then employs a graph neural ODE to model the dynamics on this graph and guides a diffusion decoder for reconstruction. A test-time adaptation strategy is adopted to dynamically update the probe topology, ensuring the model remains accurate during long-term prediction. The authors validate the accuracy and efficiency of SparseDiff on various PDE systems and real-world datasets.

**Questions:**

1. The current framework is limited to 2D regular spatial domain. Many real-world systems, such as those in climate science or sensor networks, involve data sampled from irregular or sparse spatial domains. Can you comment on the applicability of SparseDiff to such irregular or sparse data?

2. l. 143 $v_i \in R^{t\times h\times w}$. l.38  $v_i \in R^{h\times w}$. There seems to be a notational inconsistency. Can you further clarify the precise algorithm for computing the probe features?

3. How were the two hyperparameters (number of codewords $K$, and the threshold $\tau$ ) selected in practice for the experiments? Can you provide some insight into the influence of the hyperparameters on the performance?

4. It is unclear to me why the test-time adapting prediction improves the performance. Can you provide a quantitative explanation for why this re-encoding step leads to a better-aligned probe topology and improved performance? It will be good to do an ablation study that isolates the effect of adaptation. To do this, can you compare the following two models on the Navier-Stokes experiment and report the result as in Table 1?
    - SparseDiff without test-time adaptation
    - G-LED with test-time adaptation

**Ethical Concerns:**

["NO or VERY MINOR ethics concerns only"]

**Final Justification:**

I maintain my positive score.

**Limitations:**

yes

**Quality:**

3

**Strengths And Weaknesses:**

**Strengths**
- novel idea of test-time adapting prediction
- Strong performance over baselines across multiple PDE systems and a real-world dataset.
- Ablation studies on probe selection and edge weighting


**Weaknesses**

see questions

---

> ### Author Rebuttal · Authors · 2025-07-31
>
> Thank you so much for your thoughtful review and your suggestions for improvement. Below we address the concerns raised in your review:
>
> > ***Question 1: The current framework is limited to 2D regular spatial domain... Can you comment on the applicability of SparseDiff to such irregular or sparse data?***
>
> Thanks for the thoughtful question. While our current implementation operates on 2D regular grids, SparseDiff can be applied to irregular or sparse spatial data by first transforming the inputs into a regular grid through interpolation or zero-filling, enabling processing within the same framework.
>
> To validate this, we conduct two experiments:
>
> **a. Sparse sampling on Navier-Stokes system:**
> On the Navier-Stokes system, we randomly sample only 10% of grid points. These sparse samples are then interpolated onto a 128×128 regular grid before being fed into SparseDiff. Despite the sparsity, SparseDiff outperforms baselines that are given full-resolution inputs.
>
> | Method                        | RMSE × 10⁻² ↓ | SSIM × 10⁻¹ ↑ |
> | ----------------------------- | ------ | ------ |
> | FNO (full grid)               | 15.99      | 7.26     |
> | G-LED (full grid)             | 12.33     | 8.10      |
> | SparseDiff (full grid)             | 11.39     | 8.36      |
> | SparseDiff (10% observations) | 12.27  | 8.19  |
>
> **b. Irregular 2D wave equation dataset [1]:**
> We also evaluate SparseDiff on the 2D wave equation with observations sampled on irregular meshes. These inputs are first completed onto regular grids using interpolation within observed regions and zero-filling in excluded regions before being processed by the model. Compared to FNO, SparseDiff maintains stable prediction accuracy.
>
> | Method                        | RMSE × 10⁻² ↓ | SSIM × 10⁻¹ ↑ |
> | ----------------------------- | ------ | ------ |
> | FNO               | 23.76      | 7.81      |
> | SparseDiff        | 15.58      | 8.25      |
>
> While these results demonstrate practical applicability to irregular or sparse inputs, we acknowledge certain limitations of this approach:
>
> - For highly non-uniform spatial distributions, interpolating to a uniform grid can be problematic. Also, the sparse observation may be too under-sampled to allow reliable interpolation, leading to unnecessary computational cost or significant reconstruction errors.
>
> - If the distribution of spatial inputs shifts between training and testing, performance may degrade, as our current model lacks coordinate-querying capabilities (unlike operator-learning models such as neural operators), and cannot generalize across arbitrary spatial layouts.
>
> Nevertheless, the core design of SparseDiff is not inherently tied to regular grids. The key innovation of SparseDiff lies in its latent probe representation, which aggregates and models regional dynamics through sparse, learnable units. This representation is inherently flexible and does not require regular spatial structures.
>
> The main constraint stems from the UNet-based diffusion decoder, which requires regular-grid input. However, this is a design choice rather than a fundamental limitation of the method. In future work, the diffusion decoder could be replaced with architectures that naturally support irregular domains, such as Graph Neural Networks. Alternatively, it could be substituted with other super-resolution models, including transformer-based approaches and operator-learning models, to further extend SparseDiff to irregular spatial settings.
>
>
> > ***Question 2: l.143 $v_i \in \mathbb{R}^{t \times h \times w}$. l.138 $v_i \in \mathbb{R}^{h \times w}$. There seems to be a notational inconsistency...***
>
> We appreciate the reviewer’s careful attention to notation. The expression $v_i \in \mathbb{R}^{h \times w}$ was a notational error—$v_i$ is intended to denote the spatial location of the $i$-th probe, not a high-dimensional tensor.
>
> The temporal feature vector should be written as \(\mathbf{v}_i \in \mathbb{R}^l\), where \(l = 10\) is the lookback window size. The previous notation \(\mathbb{R}^{t \times h \times w}\) was a mistake. We will clarify and correct both notations in the revised version.
>
> > ***Question 3: How were the two hyperparameters (number of codewords $K$, and the threshold $\tau$) selected in practice for the experiments?...***
>
> **Choice of $K$ (Codebook Size):**
>
> The hyperparameter $K$ controls the number of discrete codewords used to represent latent dynamics and is selected based on dataset complexity.
>
> For the Navier–Stokes system which exhibits highly chaotic and multi-scale dynamics, we set $K = 150$ to ensure sufficient capacity. For the more regular Swift–Hohenberg system, a smaller size $K = 30$ can already perform well.
>
> In general, choosing $K$ involves a trade-off: larger values offer more expressiveness but may lead to under-utilized codewords and inefficiency; smaller values reduce capacity and risk underfitting.
>
> As shown in Section 4.4, Figure 5(a), performance improves with increasing $K$ up to ~150, then plateaus or slightly degrades. Overall, SparseDiff remains robust across a broad range of $K$, highlighting its flexibility in codebook sizing.
>
> **Choice of $\tau$ (Topology Update Threshold):**
>
> The threshold $\tau$ controls how sensitively the topology is updated based on latent mismatch. A smaller $\tau$ triggers more frequent updates, improving accuracy by promptly correcting structural drift, but increases computational cost. A larger $\tau$ reduces update frequency, improving efficiency but risking misalignment with evolving dynamics.
>
> In Figure 7(a), we compare the adaptive strategy (governed by $\tau$) with fixed-interval updates. The adaptive method strikes a balance:
> It achieves higher accuracy than low-frequency updates (e.g., every 5–10 steps) and lower runtime than high-frequency updates (e.g., every 1–3 steps). This shows that the threshold-based adaptation offers a practical trade-off between precision and efficiency.
>
> The table below quantifies this balance across different $\tau$ values:
>
> | $\tau$ | Avg. Update Frequency (steps) | RMSE × 10⁻² ↓ | Runtime (s) ↓ |
> | -------------- | ----------------------------- | ------ | ------------- |
> | 0.001           | 1.25                           | 8.77  | 157           |
> | 0.003           | 2.17                           | 10.03  | 114           |
> | 0.005           | 3.89                           | 11.91  | 93         |
> | 0.008           | 6.15                           | 15.43  | 79           |
> | 0.012           | 10.73                           | 25.68  | 70           |
>
> As shown, $\tau$ controls a trade-off between accuracy and efficiency and can be tuned based on task needs. The adaptive mechanism flexibly updates topology more often during rapid dynamics and less often when the system is stable, enabling efficient yet accurate long-term prediction.
>
> > ***Question 4: ...Can you provide a quantitative explanation for why this re-encoding step leads to a better-aligned probe topology and improved performance? ...can you compare the following two models on the Navier-Stokes experiment and report the result as in Table 1?***
> > - ***SparseDiff without test-time adaptation***
> > - ***G-LED with test-time adaptation***
>
> In spatiotemporal PDEs, global patterns evolve through structural transitions, such as mode aggregation, symmetry breaking, and wavefront propagation. Therefore, a fixed encoder, set at initialization, can quickly become misaligned, especially in long-term prediction. Without updates, the misaligned latent space leads to inaccuracies in prediction and decoding.
>
> This motivates our test-time adaptation strategy, which dynamically updates the topology based on emerging dynamics. Frequent updates help maintain alignment with dominant structures, ensuring that the decoder receives accurate, structure-aware conditioning. In contrast, a static topology can introduce bias and compounding errors.
>
> We further test the performance of SparseDiff without test-time adaptation using a fixed topology throughout prediction and G-LED with adaptation:
>
> - **SparseDiff w/o adaptation** suffers from severe degradation on Navier–Stokes, with RMSE exceeding 100, indicating a substantial divergence from the ground truth trajectory due to the static and increasingly misaligned probe structure.
>
> - **G-LED w/ adaptation**: results are included in Figure 7(a) (marked with cross symbols) and further summarized in the Table below. Although G-LED uses simple uniform downsampling to define its latent space, its periodic updates help capture temporal variations. As a result, it achieves moderate prediction accuracy, albeit lower than SparseDiff, and incurs higher computational cost due to the lack of adaptive locality.
>
> For reference, SparseDiff takes only **163s** at interval 1 and **73s** at interval 10, which are substantially faster than G-LED under comparable settings.
>
> | Re-encoding Interval | RMSE × 10⁻² ↓   | Runtime (s) ↓ |
> | -------------------- | -------- | ------------- |
> | 1             | 12.59 | 6950      |
> | 2          |  12.87   | 4875          |
> | 3      | 13.05     | 3537         |
> | 4      | 13.29     | 3007          |
> | 5      | 13.72     | 2642          |
> | 6      | 14.46     | 2460          |
> | 7      | 16.60     | 2214          |
> | 8      | 17.92     | 2179          |
> | 9      | 18.58    | 2003          |
> | 10      | 19.85     | 1877          |
>
>
> In conclusion, test-time adaptation is essential for PDE prediction tasks under encoding-decoding scheme. It ensures that the low-dimensional representation remains aligned with the evolving system state, preventing error accumulation and enhancing long-term prediction stability.
>
>
>
> ## Reference
> [1] Zhao, Q., Lindell, D. B., & Wetzstein, G. (2022). Learning to solve PDE-constrained inverse problems with graph networks. arXiv preprint arXiv:2206.00711.

---

> > ### Author Response · Authors · 2025-08-04
> > **Follow-up on Rebuttal and Invitation for Discussion**
> >
> > Dear Reviewer E8yK,
> >
> > We sincerely thank you for your thoughtful reviews of our submission [Submission ID: 20233].
> >
> > We have carefully addressed all raised concerns in our rebuttal, including providing additional experimental results and clarifying key methodological details.
> >
> > We kindly invite you to review our responses and engage in the discussion, should you have any further questions or feedback. Your insights are greatly appreciated and would be invaluable for a fair and comprehensive evaluation of our work.
> >
> > Thank you again for your time and contributions!
> >
> > Best regards,
> >
> > Authors

---

> > ### Comment · Reviewer_E8yK · 2025-08-05
> >
> > Thank you for your response. Your explanation has addressed most of my concerns. I will maintain my original rating of 4.

---

> > > ### Author Response · Authors · 2025-08-05
> > >
> > > Dear Reviewer E8yK,
> > >
> > > We sincerely thank you for your thoughtful consideration of our rebuttal and for your supportive assessment. Your constructive feedback has been invaluable in helping us strengthen the paper.
> > >
> > > Best regards,
> > >
> > > Authors

---

### Official Review · Reviewer_X3mc · 2025-07-02

**Clarity:** 2
**Significance:** 2
**Originality:** 2
**Rating:** 4
**Confidence:** 3

**Summary:**

This paper proposes SparseDiff, a Sparse Diffusion Autoencoder for efficient long-term prediction in spatiotemporal systems. It encodes global dynamics via a sparse set of probes using a codebook-based encoder, models them with a graph neural ODE, and reconstructs the full state with a diffusion decoder. The probe topology adapts at test time, enabling dynamic tracking. Experiments on PDE benchmarks and a climate dataset show superior accuracy and efficiency compared to state-of-the-art baselines.

**Questions:**

1. Is the topological structure present in every frame? And is it just the probe states that incorporate information from all previous frames up to the current one? If so, it would be helpful to clarify this distinction in the paper, as it’s not immediately clear whether the topology itself evolves over time or remains static while only the probe representations are temporally informed.
2. Why is the dimension of $v_i$ given as $h \times w$ ? Typically, one would expect $v_i$ to represent a vector or feature at a probe, which would usually have a fixed channel or embedding dimension. If $h\times w$ refers to spatial dimensions, this seems inconsistent with the idea of sparse probe representations. Could the authors clarify the meaning and rationale behind this shape?
3. In the first equation of Section 3.2, it’s recommended not to use $t$ as the integration variable, as it may cause confusion with the timestep index. Additionally, it's unclear whether $h_t$ and $h_i$ share the same meaning or dimensionality—does $h_t$ refer to the representation at time $t$, while $h_i$ denotes the representation of the $i$-th probe? Clarifying this distinction would help avoid ambiguity in notation.
4. In line 177, should the superscript be $N$ instead?
5. Are the Codebook Encoder and Diffusion Decoder trained separately, or is the entire model trained end-to-end?

**Ethical Concerns:**

["NO or VERY MINOR ethics concerns only"]

**Final Justification:**

My initial concerns have been addressed in the authors' response. I have therefore revised my score to 4.

**Limitations:**

See Weakness

**Quality:**

2

**Strengths And Weaknesses:**

### Strengths
1. The manuscript introduces a practical strategy for updating probe selections and topologies dynamically at test time via codebook re-encoding. This design is argued to provide an advantage for capturing emergent spatiotemporal dynamics compared to conventional static encoders and predictors.
2. The experiments are thorough and comprehensive. On PDE benchmarks and the SEVIR climate dataset, SparseDiff reduces prediction error by an average of 49.99% compared to leading baselines (Table 1), while only using about 1% of the input spatial grid. Such results suggest the framework can maintain accuracy with substantial efficiency gains. Moreover, they conduct ablation studies  to emphasize the necessity of the codebook-driven probe encoder and edge-aware attention.

### Weaknesses
1. Notation and method description are occasionally ambiguous. For example, Section 3.1.1 is overly difficult to follow, with dense explanations and somewhat confusing notation (though not incorrect). I recommend revising this section for clarity—streamlining the writing and organizing the symbols more clearly would greatly improve readability.
2. The edge weighting scheme for probe topology (Section 3.1.1) is domain-agnostic and not learned end-to-end, as acknowledged under limitations. While the ablation (Figure 6c) shows this is better than unweighted edges, it leaves open whether optimal graph connectivity could be learned or data-driven; alternatively, comparisons to other heuristic graph construction methods (e.g., k-nearest neighbors, spectral approaches) are not included.

---

> ### Author Rebuttal · Authors · 2025-07-31
>
> We deeply appreciate the time you took to review our work and your meticulous comments for improvement. Below we address the concerns raised in your review:
>
> > ***Question 1: Is the topological structure present in every frame? And is it just the probe states that incorporate information from all previous frames up to the current one?...***
>
> We appreciate the reviewer’s question. The probe topology is not fixed throughout prediction. Instead, it is periodically updated during test-time via re-encoding. Specifically, the prediction horizon is divided into adaptive sub-windows. At the start of each sub-window, a new probe graph is constructed using the encoder and codebook.
>
> In contrast, probe states are continuously updated at each prediction step, as their trajectories evolve over time. These changes reflect the system’s ongoing dynamics even when the underlying graph remains fixed within a window. The topology is updated only when a latent mismatch is detected between current probe states and their assigned codewords. As detailed in **Section 3.3**, this mismatch is quantified by the latent consistency score $\chi_t$, which measures how well the current probe state aligns with its nearest codeword in latent space. When $\chi_t$ falls below a threshold $\tau$, the system triggers re-encoding and reconstructs a new topology, allowing adaptive alignment with evolving spatiotemporal patterns (see Section 3.3 for implementation details and Figure 4 for illustration).
>
> Additionally, probe states are not derived from all previous frames but from a fixed-length lookback window. As noted in **Appendix B**, we use a 10-frame history in all experiments to balance memory efficiency and temporal context.
>
> We will revise the paper to clarify these distinctions.
>
>
> > ***Question 2: Why is the dimension of $v_i$ *given as* $h \times w$? Typically, one would expect $v_i$ to represent a vector or feature at a probe, which would usually have a fixed channel or embedding dimension. If $h \times w$ refers to spatial dimensions, this seems inconsistent with the idea of sparse probe representations...***
>
>
> We appreciate the reviewer’s observation. The expression $v_i \in \mathbb{R}^{h \times w}$ was a notational error. Here, $v_i$ is intended to indicate the spatial location of the $i$-th probe, not a high-dimensional tensor.
>
> The actual temporal feature vector associated with each probe is denoted as $\mathbf{v}_i \in \mathbb{R}^l$, where $l$ is the lookback window size (10 in our experiments). This will be clarified and corrected in the revised version.
>
> > ***Question 3: In the first equation of Section 3.2, it’s recommended not to use* $t$ as the integration variable... Additionally, it’s unclear whether $h_t$ and $h_i$ share the same meaning or dimensionality—does $h_t$ refer to the representation at time $t$, while $h_i$ denotes the representation of the $i$-th probe?...***
>
> We thank the reviewer for the helpful suggestion. We agree that using $t$ as the integration variable in Section 3.2 may cause confusion, and we will revise it to a different symbol (e.g., $\tau$) to avoid overlap with the timestep index.
>
> The reviewer is also correct that $h_t$ refers to the representation at time $t$, while $h_i$ denotes the representation of the $i$-th probe. We chose not to write $h_i^t$ uniformly because different equations emphasize different aspects: some focus on temporal evolution without distinguishing probes, while others highlight probe interactions without indexing time. We will clarify this in the final version.
>
> > ***Question 4: In line 177, should the superscript be N instead?***
>
> Yes, the superscript should indeed be $N$ rather than $n$. We will correct it in the final version.
>
>
> > ***Question 5: Are the Codebook Encoder and Diffusion Decoder trained separately, or is the entire model trained end-to-end?***
>
> We appreciate the reviewer’s question. The codebook-based Sparse Encoder, probe-graph diffusive predictor, and guided Diffusion Decoder are trained as separate modules.
>
> End-to-end training is not feasible in our framework because the probe selection and graph construction processes involve discrete operations (e.g., nearest-neighbor lookup, subgraph assignment) that block gradient flow. Additionally, during the diffusion decoding stage, we apply explicit guidance (Equation 3) by injecting probe values into the denoising process step-by-step, rather than relying on implicit conditioning. This guidance mechanism is inherently non-differentiable, further limiting the feasibility of end-to-end optimization.
>
> We experimented with combining both implicit and explicit conditioning for the diffusion model. However, adding implicit conditioning significantly increased the number of trainable parameters without providing obvious performance gains. This supports our decision to rely on explicit guidance for accurate reconstruction.
>
> | Model | Parameters (M) | RMSE × 10⁻² ↓ |
> | --------------------- | ---------- | ------------- |
> | Ours | 26  | 2.912 |
> | Implicit conditioning | 74  | 2.809         |
>
> While the model is not end-to-end trainable, modular training offers several practical advantages. It allows each component to be developed, pre-trained, or replaced independently, facilitates better interpretability, and reduces the memory and compute requirements during training. This modularity also makes it easier to adapt the framework to new physical systems or plug in more advanced predictors and decoders in future work.
>
> > ***Weakness 1: ...Section 3.1.1 is overly difficult to follow, with dense explanations and somewhat confusing notation (though not incorrect). I recommend revising this section for clarity...***
>
> We appreciate the reviewer’s feedback regarding clarity and notation in Section 3.1.1. We agree that this section can be better structured and will revise it to improve readability. We hope that our responses to Q2–Q4 above have helped clarify some of the core notational choices and methodological distinctions. We will incorporate these clarifications into the paper to reduce potential confusion.
>
> > ***Weakness 2: The edge weighting scheme for probe topology (Section 3.1.1) is domain-agnostic and not learned end-to-end... alternatively, comparisons to other heuristic graph construction methods (e.g., k-nearest neighbors, spectral approaches) are not included.***
>
> We appreciate the reviewer’s comments regarding the edge weighting scheme in our probe topology. Our edge weighting scheme is **partially heuristic, partially learnable**, and the resulting topology is **fully differentiable** during training. While the graph structure is not constructed end-to-end due to the discrete nature of probe grouping, the diffusion weights used by the predictor are trainable, allowing data-driven adaptation over a spatially informed backbone.
>
>
> Specifically, our predictor is based on Graph Neural Diffusion (GRAND) [1], which models information propagation on a graph as a continuous diffusion process:
>
>
> $$
> h_t = h_0 + \int_0^t A  h_\tau \, d\tau,
> $$
>
> where $A$ defines the diffusion strength, and each $a_{ij}$ encodes how much node $j$ contributes to node $i$’s evolution over virtual time.
>
> In the original GRAND framework, diffusivity $ a_{ij} $ is computed via attention over learnable probe features. In our case, each node represents a region-aggregated embedding constructed by averaging features of all grid points within the codeword region $ u_{c_i} $. Therefore, probe similarity should reflect not only feature proximity but also region-level spatial adjacency.
>
> To reflect this, we introduce an additional edge weight term $e_{ij}$, which quantifies the degree of spatial adjacency between probe regions. Specifically, $e_{ij}$ measures the fraction of non-internal neighboring points in $u_{c_i}$ that are adjacent to points in $u_{c_j}$. This term is integrated into the attention-based diffusivity calculation as:
> $$
> a(h_i, h_j) = \text{softmax}\left( \frac{(W_K h_i)^T W_Q h_j + W_E e_{ij}}{d_k} \right),
> $$
>
> where $W_E$ is a learnable weight matrix. This allows $e_{ij}$ to serve as an additional input feature rather than a fixed multiplicative factor, making the final diffusion strength $a_{ij}$ still trainable via backpropagation.
>
> Moreover, we do not predefine a fixed graph. Instead, we assume full connectivity and let the attention-based matrix $A$ learn effective interactions. When $a_{ij} \approx 0 $, it implies no effective connection, enabling the topology to adapt during training.
>
> To assess our edge weighting scheme, we compare against:
>
> a. Graph Kernel Network (GKN): A learnable graph construction method [2];
>
> b. Heuristic alternatives using the same GRAND predictor but replacing $e_{ij}$ with:
> - k-nearest neighbors based on Euclidean coordinates ($k = 20\% $) of probe count);
> - Spectral clustering followed by KNN in embedding space.
>
>
> Results are as follows (average RMSE over 100-step autoregressive prediction on LO system):
>
> | Method                                 | RMSE × 10⁻² |
> |----------------------------------------|--------|
> | Ours    |    2.873    |
> | GKN      |   4.210       |
> | GRAND + kNN           |   3.287     |
> | GRAND + spectral       |   3.019     |
>
>
> As shown, our region-aware edge scheme yields better performance than both heuristic variants and learnable graph construction methods, demonstrating that encoding region-level adjacency improves the expressiveness of probe-based dynamics modeling.
>
> ## Reference
> [1] Chamberlain, B. P., Rowbottom, et. al. (2021). GRAND: Graph neural diffusion. arXiv preprint arXiv:2106.10934.
>
> [2] Li, Z., Kovachki, N., Azizzadenesheli, et. al. (2020). Neural operator: Graph kernel network for partial differential equations. arXiv preprint arXiv:2003.03485.

---

> > ### Comment · Reviewer_X3mc · 2025-08-05
> >
> > I appreciate the authors’ detailed responses and revisions, which have addressed most of my concerns. However, I still think the proposed separate training approach to be somewhat inflexible, and the technical exposition of the method, as noted in my initial review (Weakness 1), remains a limitation. Given these persisting issues, I maintain my original assessment.

---

> ### Author Response · Authors · 2025-08-04
> **Follow-up on Rebuttal and Invitation for Discussion**
>
> Dear Reviewer X3mc,
>
> We sincerely thank you for your thoughtful reviews of our submission [Submission ID: 20233].
>
> We have carefully addressed all raised concerns in our rebuttal, including providing additional experimental results and clarifying key methodological details.
>
> We kindly invite you to review our responses and engage in the discussion, should you have any further questions or feedback. Your insights are greatly appreciated and would be invaluable for a fair and comprehensive evaluation of our work.
>
> Thank you again for your time and contributions!
>
> Best regards,
>
> Authors

---

> ### Author Response · Authors · 2025-08-08
>
> Dear reviewer X3mc,
>
> Thank you for your thoughtful comments. We would like to further clarify our approach.
>
> Our pipeline **can be trained end-to-end for downstream prediction tasks once the graph structure is obtained.** The graph serves as an external, task-agnostic knowledge representation, independent of any specific prediction objective. Therefore, it does not require end-to-end optimization, while the subsequent modules—probe encoding, prediction, and diffusion decoding—are indeed suitable for such training.
>
> More specifically, our model consists of three components: the sparse encoder, predictor, and diffusion decoder. The sparse encoder extracts key topological structures that capture the system's evolution, yielding an essential graph representation. This graph functions as a **structural inductive bias** [1], guiding the model toward solutions consistent with the system’s underlying spatiotemporal dynamics. Similar concepts exist in other domains: the governing equations embedded in Physics-Informed Neural Networks (PINNs) [2], Hamiltonian priors in Hamiltonian Neural Networks [3], or the translation invariance bias in Convolutional Neural Networks (CNNs) for image analysis [4]. While the graph itself is **task-agnostic** and thus **does not require joint end-to-end optimization**, the sparse encoder also includes the probe encoding module, whose learned features are task-dependent and can be further refined in an end-to-end training process.
>
> In our original design, however, the predictor and diffusion decoder were trained separately rather than end-to-end, as direct end-to-end training proved 2.1× slower than the separate-training scheme and prone to instability.
>
> Based on your insightful comment, we carefully redesign an improved workflow to **make end-to-end optimization more practical.** First, we train the individual components separately to obtain stable intermediate representations. Then, we fine-tune the full model end-to-end using a conditional diffusion model, in which the predictor’s output serves as the condition embedding for the diffusion decoder. This setup enables joint optimization of prediction and reconstruction losses, allowing effective gradient flow across modules.
>
> We conduct an experiment on the Real Climate dataset. We find that the fine-tuned model outperformed the non-end-to-end version, with **notable improvements in noise robustness**. The results are summarized in the table below (SSIM × 10⁻¹ ↑):
>
> | Noise Strength (%) | End-to-End | Separate |
> | ------------------ | ---------- | -------- |
> | 0.0                | 7.793      | 7.725    |
> | 0.2                | 7.735      | 7.693    |
> | 0.4                | 7.699      | 7.618    |
> | 0.6                | 7.603      | 7.502    |
> | 0.8                | 7.439      | 7.241    |
> | 1.0                | 7.295      | 7.017    |
>
> Therefore, end-to-end training is viable and particularly advantageous in enhancing noise robustness. However, it involves a trade-off between precision and efficiency. For applications prioritizing training efficiency, the unconditional diffusion setting offers shorter training time with acceptable accuracy. Conversely, when robustness to noise is critical, end-to-end training becomes the preferable choice.
>
> Once again, we sincerely appreciate your helpful suggestions, which have led to significant refinements in our approach. We will include this adjustment in the camera-ready paper.
>
> Best regards,
>
> Authors
>
> ## Reference
> [1] Battaglia, P. W., Hamrick, J. B., Bapst, V., Sanchez-Gonzalez, A., Zambaldi, V., Malinowski, M., ... & Pascanu, R. (2018). Relational inductive biases, deep learning, and graph networks. arXiv preprint arXiv:1806.01261.
>
> [2] Raissi, M., Perdikaris, P., & Karniadakis, G. E. (2019). Physics-informed neural networks: A deep learning framework for solving forward and inverse problems involving nonlinear partial differential equations. Journal of Computational physics, 378, 686-707.
>
> [3] Greydanus, S., Dzamba, M., & Yosinski, J. (2019). Hamiltonian neural networks. Advances in neural information processing systems, 32.
>
> [4] LeCun, Y., & Bengio, Y. (1998). Convolutional networks for images, speech, and time series. The handbook of brain theory and neural networks.

---

### Official Review · Reviewer_sBsw · 2025-07-02

**Clarity:** 2
**Significance:** 3
**Originality:** 3
**Rating:** 4
**Confidence:** 2

**Summary:**

This paper proposes SparseDiff, a novel framework for long-term prediction of complex spatiotemporal systems. SparseDiff uses a codebook-based sparse encoder to construct a “probe topology” capturing low-dimensional spatial structures, and a guided diffusion model to reconstruct full-domain states. The key innovation lies in the test-time adaptation mechanism that dynamically updates the encoding based on the current system state. The approach is validated on both synthetic PDE systems and real-world climate data, with strong quantitative and qualitative performance.

**Questions:**

-	In Section 3.1.2, you highlight the differences between your method and previous works [14, 13, 12], but these methods are not included in the experimental comparisons. Could you clarify why they were omitted?

-	How is the value of k, described in Section 3.1.1, chosen, and how does it affect the final performance and effiency?

-	The trainable parameter of SparseDiff are larger than all of other baseline, could you also compare  training GPU hours for SparseDiff and other baselines?


## Typos:

-	In line 101, \epolsion should be \epolsion_n? Keep consistent with Equation (2), where the subscript denotes the specific noise term at step n.

-	Figure 7(b) lacks explanation; it is unclear what the term 'Rollout step' refers to.

**Ethical Concerns:**

["NO or VERY MINOR ethics concerns only"]

**Final Justification:**

I will maintain my positive score!

**Limitations:**

Yes.

**Paper Formatting Concerns:**

No.

**Quality:**

3

**Strengths And Weaknesses:**

## Strengths

- The Test-Time Adaptation  mechanism based on latent similarity offers a practical solution for capturing emerging spatiotemporal structures, which help improve long-term prediction accuracy and inference efficiency.

- Extensive experiments on diverse PDE benchmarks and real-world data demonstrate clear advantages over baselines, including FNO and G-LED.

- The method that uses sparse probes to construct a reduced-order model is very interesting. However, I'm not very familiar with this field, so I'm not sure about its novelty.

## Weeknesses :

- The number of trainable parameters in SparseDiff is larger than that of all other baselines, and the training efficiency compared to the other methods is not clearly discussed.

- The experimental configurations for the 'Robustness,' 'Generalization,' and 'Trade-off between Accuracy & Efficiency' sections are not consistent. For example, the Robustness experiment compares against G-LED on the Navier-Stokes system, while the Generalization experiment compares against FNO on the Cylinder Flow system. Could you clarify the rationale behind these choices, or consider adding experiments on additional PDE systems with more consistent baseline comparisons?

---

> ### Author Rebuttal · Authors · 2025-07-31
>
> We sincerely thank you for your thoughtful and constructive feedback! We truly appreciate the time and effort you devoted to reading our paper and offering valuable insights.
>
> > ***Weakness 1 & Question 3: ...the training efficiency compared to the other methods is not clearly discussed; ...could you also compare training GPU hours for SparseDiff and other baselines?***
>
> We sincerely thank the reviewer for raising this important point. After the initial submission, we revisited the model complexity of SparseDiff and experimented with a smaller version containing ~**25.8M** parameters, which is of similar scale to G-LED and FNO.
>
> Notably, this lighter model still achieves competitive or even better performance across various systems (RMSE × 10⁻²↓ ):
>
> | System      | 25.8M SparseDiff   | 55M SparseDiff    | G-LED|
> |-------------|--------------------|-------------------|--------------------|
> | LO          | 2.912 ± 0.187      |2.873 ± 0.237     |6.506 ± 0.395      |
> | NS          | 11.130 ± 3.241     |11.390 ± 5.359     |12.334 ± 0.485     |
> | SH          | 7.628 ± 3.102      |7.700 ± 2.540      |8.214 ± 0.937      |
> | CY          | 8.544 ± 0.239      |8.012 ± 0.655      |10.021 ± 0.873      |
> | Real-world  | 7.957 ± 1.207      |7.742 ± 2.237      |10.304 ± 2.009      |
>
> Interestingly, on the NS and SH systems, the smaller model outperforms the original 50M-parameter version. We suspect this is due to improved optimization dynamics and stronger generalization.
>
> Regarding training efficiency, we conducted timing benchmarks on a single A100 GPU:
>
> - SparseDiff: ~500 minutes (400 epochs to converge)
> - G-LED: ~3100 minutes (400 epochs to converge)
> - FNO: ~30 minutes (20 epochs to converge)
>
> While diffusion-based models typically require more epochs to converge, SparseDiff is much more efficient than G-LED in both training and inference. This efficiency results from its sparse design and lightweight predictor.
>
> We will include these comparisons in the final version to emphasize that SparseDiff’s strong performance is due to its structural advantages and effective design, not just a larger parameter count.
>
> > ***Weakness 2: The experimental configurations for the 'Robustness,' 'Generalization,' and 'Trade-off between Accuracy & Efficiency' sections are not consistent...***
>
> We intentionally chose different PDE systems across sub-experiments to show that SparseDiff performs robustly across various settings, not just a single task.
>
> For baseline selection, we compared against the **strongest-performing** method on each system to highlight SparseDiff’s advantages. For example:
>
> * On Navier–Stokes, G-LED performs best among baselines, so we use it for robustness experiments.
> * On Cylinder Flow, FNO has the highest SSIM, making it the most competitive baseline for generalization tests.
>
> To improve consistency, we add the following experiments:
>
> **(a) Robustness to codebook size:**
> In the robustness experiment on codebook size (Fig. 4a), we compare against G-LED on the Navier–Stokes system and report FNO’s performance, which yields an SSIM of 0.726. This confirms that SparseDiff outperforms multiple baselines when the codebook size is properly chosen and remains robust across various sizes.
>
> **(b) Robustness to noise:**
> We report SparseDiff’s performance on the Navier–Stokes system with noisy inputs, in addition to results on a real-world system:
>
> | Noise Strength (%) | SSIM × 10⁻¹↑    |
> |-------------|--------------------|
> | 0.0          | 8.364 ± 0.992      |
> | 0.2        | 8.299 ± 0.328     |
> | 0.4        | 8.181 ± 0.623      |
> | 0.6        | 8.092 ± 0.837      |
> | 0.8        | 7.907 ± 0.477      |
> | 1.0        | 7.710 ± 1.402      |
>
> This demonstrates that SparseDiff maintains strong performance under noise, with SSIM only slightly below G-LED’s when the noise exceeds 0.6%. This demonstrates the robustness of our model to input perturbations.
>
> **(c) Generalization:**
> We evaluate SparseDiff’s generalization on the Navier–Stokes system, extending beyond the Cylinder-Flow system. The generalization parameter is the coefficient of the external forcing term. During training, the model is exposed to forcing coefficients within the in-distribution (ID) range $[0, \pi]$, while out-of-distribution (OOD) testing uses coefficients in $(\pi, 2\pi]$. We use SSIM (×10⁻¹ ↑) as the evaluation metric, and the results are summarized below:
>
> | data distribution | SparseDiff    |G-LED|
> |-------------|--------------------|--------------------|
> | ID         | 8.173 ± 0.491      | 7.945 ± 0.832     |
> | OOD        | 8.002 ± 1.197     | 7.153 ± 1.970      |
>
>
> This demonstrates that SparseDiff maintains strong performance under noise, with SSIM only slightly below G-LED’s when the noise exceeds 0.6%. This demonstrates the robustness of our model to input perturbations.
>
> > ***Question 1: In Section 3.1.2, you highlight the differences between your method and previous works [14, 13, 12], but these methods are not included in the experimental comparisons...***
>
> The cited works [12, 13, 14] explore using sparse observations as conditioning inputs to guide diffusion models, aiming to reconstruct or complete solutions from partial measurements.
>
> For instance, S³GM [13] focuses on recovering the full spatiotemporal field from sparse sensor data, while DiffusionPDE [14] addresses inverse problems like solving for unknown inputs from limited observations. [12] compares different conditioning strategies in diffusion frameworks for sparse-observation settings.
>
> In contrast, our method doesn't rely on external sparse measurements but learns to extract sparse internal representations (key probes) that capture essential dynamics for prediction. This makes our approach fundamentally different.
>
> However, we agree these works are relevant and we conduct a direct comparison with DiffusionPDE [14] on the SH dataset, as it can serve as a representative example of the three methos applying diffusion models under sparse condition. The results are as follows (SSIM ×10⁻¹ ↑):
>
> | Predict Step | SparseDiff     | DiffusionPDE     |
> |-------------|--------------------| -----|
> | 10          | 9.87      | 9.52     |
> | 20        | 9.83    | 9.21  |
> | 40        | 9.80      |  8.70    |
>
> We also compare inference time (s):
>
> | Predict Step | SparseDiff     | DiffusionPDE     |
> |-------------|--------------------| -----|
> | 10          |   20    | 3000     |
> | 20        | 22    | 6000  |
> | 40        | 26      |  12000    |
>
> As shown, SparseDiff significantly outperforms DiffusionPDE in both accuracy and inference efficiency, especially in long-term prediction. This aligns with our assessment that DiffusionPDE is better suited for solving inverse or sparse-observation tasks, rather than for long-term predicting.
>
>
> > ***Question 2: How is the value of $K$, described in Section 3.1.1, chosen, and how does it affect the final performance and effiency?***
>
> The $K$ mentioned in Section 3.1.1 refers to the codebook size, a tunable hyperparameter that determines how many codewords are available to represent system dynamics.
>
> The optimal $K$ value varies across datasets depending on the complexity of the underlying patterns. For example:
>
> - On NS system, which exhibits highly complex dynamics, we use $K$ = 150 to ensure sufficient expressiveness.
>
> - On SH system, the patterns are simpler, and a smaller codebook ($K$ = 30) already achieves strong performance.
>
> In general, the choice of $K$ involves a trade-off between reconstruction quality and codebook utilization:
>
> A larger $K$ offers greater representational capacity, potentially improving accuracy. However, an excessively large $K$ may lead to many unused codewords, reducing utilization efficiency, increasing memory overhead, and even slightly degrading performance due to undertraining of rarely selected codes.
>
> The result is illustrated in Section 4.4, Figure 5(a), where we vary the codebook size $K$ on the NS system. The results show that performance plateaus around $K$ = 150, and further increasing $K$ leads to a slight performance drop—highlighting the importance of dataset-specific tuning. Notably, the plateau spans a fairly wide range of $K$ values, indicating that the method is robust to the choice of codebook size within a reasonable interval.
>
> > ***Typo Corrections:***
>
> We thank the reviewer for catching these issues:
>
> - We have corrected the typo in line 101: `\epolsion` is now consistently written as `\epolsion_n` to match the notation in Equation (2), where the subscript denotes the specific noise term at step \( n \).
>
> - The term *"Rollout step"* in the caption of Figure 7(b) refers to the time index in the autoregressive prediction sequence, where each step corresponds to one forward prediction in the rollout process.
>
> > ***Strength 3: ...is very interesting ... not sure about its novelty.***
>
> Thank you for your positive feedback. The sparse probe scheme indeed represents a breakthrough compared to traditional encoding methods [1, 2]. Its novelty and practical utility have been recognized by multiple reviewers. For example, E8yK highlighted the novel idea of test-time adapting prediction, and X3mc pointed out its advantages in capturing emergent spatiotemporal dynamics.
>
> ## Reference
>
> [1] Yu, D., Li, X., Ye, Y., Zhang, B., Luo, C., Dai, K., ... & Chen, X. (2024). Diffcast: A unified framework via residual diffusion for precipitation nowcasting. In Proceedings of the IEEE/CVF Conference on Computer Vision and Pattern Recognition (pp. 27758-27767).
>
> [2] Gao, H., Kaltenbach, S., & Koumoutsakos, P. (2024). Generative learning for forecasting the dynamics of high-dimensional complex systems. Nature Communications, 15(1), 8904.

---

> > ### Comment · Reviewer_sBsw · 2025-08-07
> >
> > Thanks to the authors for the detailed response. All of my concerns have been addressed, and I will maintain my positive score.

---

> > > ### Author Response · Authors · 2025-08-08
> > >
> > > Dear reviewer sBsw,
> > >
> > > We are glad that our responses have addressed all of your concerns. We sincerely appreciate your positive feedback and continued support of our work.
> > >
> > > Best regards,
> > >
> > > Authors

---

> ### Author Response · Authors · 2025-08-04
> **Follow-up on Rebuttal and Invitation for Discussion**
>
> Dear Reviewer sBsw,
>
> We sincerely thank you for your thoughtful reviews of our submission [Submission ID: 20233].
>
> We have carefully addressed all raised concerns in our rebuttal, including providing additional experimental results and clarifying key methodological details.
>
> We kindly invite you to review our responses and engage in the discussion, should you have any further questions or feedback. Your insights are greatly appreciated and would be invaluable for a fair and comprehensive evaluation of our work.
>
> Thank you again for your time and contributions!
>
> Best regards,
>
> Authors

---

> > ### Comment · Area_Chair_W3aw · 2025-08-05
> >
> > Dear Reviewer, as the deadline for this key phase of the NeurIPS review process is just a few days away, we’d greatly appreciate your engagement in any remaining discussions with the authors.

---

### Author Response · Authors · 2025-08-09

# General Comments by Authors

We sincerely thank all the reviewers for their insightful reviews and constructive feedback, which have been invaluable in helping us refine and strengthen our paper.

The reviewers generally recognized the significance of our work. They acknowledged that our proposed test-time adaptation is a **"novel idea"** (E8yK) and a **"practical strategy"** (sBsw, X3mc) for a challenging problem. They highlighted the model's **"state-of-the-art long-horizon accuracy"** (WcE9) and its **"strong performance"** (E8yK) demonstrated through **"extensive"** (sBsw) and **"thorough"** (X3mc) experiments on diverse benchmarks.

The reviewers also raised several important and constructive concerns. We have made our best effort to address all points by providing detailed clarifications, theoretical justifications, and extensive new experimental results. Below is a summary of the key issues discussed and resolved during the rebuttal phase.

- **Model Complexity and Training Efficiency:**

  In response to **Reviewer sBsw**, we demonstrated that a smaller version of SparseDiff (~25M parameters) remains highly competitive, proving that its strong performance is due to its structural design, not just parameter count. We also provided detailed training time comparisons, highlighting our model's efficiency relative to other diffusion-based methods like G-LED. This resolved the reviewer's concerns.

- **Experimental Robustness and Consistency:**

  We addressed concerns from **Reviewer sBsw** regarding the consistency of experimental setups by adding new experiments to provide more direct comparisons. We further conducted new experiments to validate the model's robustness to input noise and its generalization capabilities on different PDE systems, addressing key questions from both **Reviewer sBsw** and **Reviewer E8yK**.

- **Impact and Necessity of Test-Time Adaptation:**

  At the request of **Reviewer E8yK**, we provided a detailed ablation study comparing SparseDiff with and without adaptation, and also applied our adaptation strategy to a representative baseline (G-LED). The results quantitatively demonstrated that our adaptive re-encoding mechanism is essential for accurate long-term prediction and is a key advantage of our framework. This was acknowledged by **Reviewer E8yK**.

- **Methodological Justification and Clarity:**

  We addressed concerns from **Reviewers sBsw, X3mc, and WcE9** by clarifying our notation, the definition of "sparsity," and the rationale behind our graph construction. In response to **Reviewer X3mc** and **WcE9**, we provided additional experiments comparing our partially learnable, physics-informed edge weighting scheme against both heuristic (kNN, spectral) and fully learnable (GKN) alternatives, demonstrating its superior performance.

- **Applicability to irregular or sparse domains:**

  In response to **Reviewer E8yK**, we demonstrated SparseDiff’s applicability by interpolating irregular/sparse inputs to regular grids, and reported performance on sparse-sampled Navier–Stokes and irregular 2D wave datasets. We also discussed limitations and potential extensions to irregular-domain decoders. The reviewer confirmed that the concerns were addressed.

- **End-to-End Training Feasibility:**

  **Reviewer X3mc** raised an excellent point about the inflexibility of separate training. Prompted by this feedback, we carefully redesigned the workflow and implemented an end-to-end fine-tuning strategy. Concretely, we first trained the predictor and diffusion decoder separately to ensure stability, and then fine-tuned them jointly using a conditional diffusion model so that prediction and reconstruction losses could be optimized together. New experimental results show that this approach is not only viable but also significantly improves the model's robustness. However, it comes with a trade-off between precision and efficiency: while end-to-end training enhances robustness, it requires longer training time compared to the unconditional diffusion setting. Overall, this represents a substantial refinement of our method.

We are very grateful for the active and valuable engagement from the reviewers. **Reviewers sBsw** and **E8yK** have both confirmed that their concerns were addressed and have maintained their positive ratings. In response to **Reviewer X3mc's** final comment, we conducted significant new work on an end-to-end training scheme, which we believe resolves their primary remaining concern and has materially improved our paper. Although **Reviewer WcE9** has not yet engaged in discussion, we believe our detailed rebuttal and the new experiments on theoretical justification, the role of sparsity, and the physical interpretation of our learned operator have thoroughly addressed all points raised in their review.

The reviewers' suggestions have been instrumental in improving our work. We remain available and eager to answer any further questions.

---

### Decision · Program_Chairs · 2025-09-17

**Decision:**

Accept (poster)

**Comment:**

Summary: This paper proposes SparseDiff, an adaptive model for spatiotemporal forecasting that dynamically updates a sparse graph representation to capture emergent patterns. Using a graph neural ODE and diffusion decoder, it reduces prediction error by 49.99% compared to baselines with a 99% reduction in spatial resolution.

Strengths: 1. The test-time adaptation mechanism provides a practical approach for capturing emerging spatiotemporal structures. 2. Extensive experiments across diverse PDE benchmarks and real-world data show clear advantages over baseline methods. 3. The technique of using sparse probes to construct a reduced-order model is highlighted as particularly interesting.

Weaknesses: 1. The paper suffers from poor writing quality and occasionally ambiguous descriptions of notation and methods. 2. Inconsistencies are noted in the experimental configurations. 3. The paper's theoretical contributions are considered overclaimed. 4.The edge weighting scheme for the probe topology is criticized for being domain-agnostic and not learned in an end-to-end manner. 5. The proposed framework is limited in applicability to 2D regular spatial domains.

During the rebuttal, the authors adequately addressed the major concerns. Reviewer X3mc acknowledged that the training pipeline, while not fully end-to-end, is a reasonable and effective approach. The sole negative score came from Reviewer WcE9, who raised a valid concern regarding the lack of a clear explanation for how prediction accuracy evolves over time (t)—a key point in the paper's theoretical claims. However, as Reviewer WcE9 did not engage in the rebuttal discussion, and based on the authors' proposed response, I believe this weakness can be tractably addressed by modifying the relevant statements without diminishing the paper's core novelty and interestingness. I therefore recommend borderline acceptance, strongly urging the authors to integrate all suggestions into their final version.